# Experience-dependent flexibility in a molecularly diverse central-to-peripheral auditory feedback system

**Michelle M Frank[1], Austen A Sitko[1], Kirupa Suthakar[2], Lester Torres Cadenas[2], Mackenzie Hunt[1], Mary Caroline Yuk[1], Catherine JC Weisz[2], Lisa V Goodrich[1]\***

[1]Department of Neurobiology, Harvard Medical School, Boston, United States; [2]Section on Neuronal Circuitry, National Institute on Deafness and Other Communication Disorders, Bethesda, United States

**Abstract** Brainstem olivocochlear neurons (OCNs) modulate the earliest stages of auditory processing through feedback projections to the cochlea and have been shown to influence hearing and protect the ear from sound-induced damage. Here, we used single-nucleus sequencing, anatomical reconstructions, and electrophysiology to characterize murine OCNs during postnatal development, in mature animals, and after sound exposure. We identified markers for known medial (MOC) and lateral (LOC) OCN subtypes, and show that they express distinct cohorts of physiologically relevant genes that change over development. In addition, we discovered a neuropeptide-enriched LOC subtype that produces Neuropeptide Y along with other neurotransmitters. Throughout the cochlea, both LOC subtypes extend arborizations over wide frequency domains. Moreover, LOC neuropeptide expression is strongly upregulated days after acoustic trauma, potentially providing a sustained protective signal to the cochlea. OCNs are therefore poised to have diffuse, dynamic effects on early auditory processing over timescales ranging from milliseconds to days.

## Editor's evaluation

This paper provides a detailed cellular and molecular characterization of the olivocochlear efferents that project to the inner ear. These specialized motoneurons are the only source of feedback from the brain to the ear and have been difficult to access. This study comprehensively categorizes the efferents, using single nucleus RNA-sequencing and 3D reconstructions of individual fibers and their pre-synaptic contacts onto target neurons in the cochlea.

\*For correspondence:
Lisa_Goodrich@hms.harvard.edu

**Competing interest:** The authors declare that no competing interests exist.

## Introduction

All sensory systems are modulated by feedback circuits that dynamically tune sensory information in response to both external experience and internal state. In the sense of vision, for instance, central efferent pathways mediate the reflexive restriction of the pupil in response to bright light and also modulate pupil diameter with changes in arousal state (*McGinley et al., 2015*). Analogously, a small group of several hundred olivocochlear neurons (OCNs) in the superior olivary complex (SOC) extend projections all the way to the peripheral hearing organ, the cochlea, enabling direct regulation of initial detection and encoding of auditory stimuli (*Rasmussen, 1946*; *Rasmussen, 1953*; *Figure 1A and B*). By integrating information from both ascending auditory pathways and descending central circuits, OCNs serve as a conduit for an animal's experiences and needs to modulate sound information at the earliest stages of sensory processing.

**eLife digest** Just as our pupils dilate or shrink depending on the amount of light available to our eyes, our ears adjust their sensitivity based on the sound environment we encounter. Evidence suggests that a group of cells known as olivocochlear neurons (OCNs for short) may be involved in this process. These cells are located in the brainstem but project into the cochlea, the inner ear structure that converts sound waves into the electrical impulses relayed to the brain. OCNs may mediate how sounds are detected and encoded "at the source."

Historically, OCNs have been divided into two groups (medial or lateral OCNs) based on different morphologies and roles in hearing. For instance, medial OCNs are thought to protect our ears against loud sounds by sending molecular signals to the inner ear cells that amplify certain auditory signals. However, it remains difficult to disentangle the precise function of the different types of OCNs, in part because scientists still lack markers that would allow them to distinguish between medial and lateral cells simply based on genetic activity.

Frank et al. aimed to eliminate this bottleneck by identifying which genes were switched on and to what degree in individual mouse medial and lateral OCNs; this was done throughout development and after exposure to loud noises. The experiments uncovered a range of genetic markers for medial and lateral OCNs, showing that these cells switch on different sets of genes relevant to their role over development. This gene expression data also revealed that two distinct groups of lateral OCNs exist, one of which is characterised by the production of large amounts of neuropeptides, a type of chemical messenger that can modulate neural circuit activity.

Further work in both developing and adult mice showed that this production is shaped by the activity of the cells, with the neuropeptide levels increasing when the animals are exposed to damaging levels of noise. This change lasts for several days, suggesting that such an experience can have long-lasting effects on how the brain provides feedback to the ear.

Overall, the results by Frank et al. will help to better identify and characterize the different types of OCNs and the role that they have in hearing. By uncovering the chemical messengers that mediate the response to loud noises, this research may contribute to a better understanding of how to prevent or reduce hearing loss.

The impact of OCNs on cochlear function is poorly understood, as few markers for these cells have been identified and the population is difficult to target either genetically or physiologically. OCNs express a variety of neurotransmitters (*Eybalin, 1993*; *Kitcher et al., 2022*; *Sewell, 2011*) and can even change their neurotransmitter profiles in response to intense sounds (*Niu and Canlon, 2002*; *Wu et al., 2020*), making it difficult to identify and catalog cell types. Developmentally and evolutionarily, OCNs are closely related to the facial branchial motor neurons (FMNs; *Frank and Goodrich, 2018*; *Fritzsch and Elliott, 2017*). Both populations derive from common motor neuron progenitors and share expression of motor neuron markers such as Islet-1 (*Frank and Goodrich, 2018*; *Fritzsch and Elliott, 2017*), although OCNs are not considered motor neurons. Expression of the zinc-finger transcription factor GATA3 seems to be crucial for specifying an OCN fate, but little else is known about how OCNs and FMNs diverge, or how OCN subtypes differentiate from one another and acquire their specialized attributes (*Pata et al., 1999*; *Karis et al., 2001*). Thus, identifying markers that can distinguish OCN subtypes from one another and from neighboring FMNs is critical for understanding the anatomical and molecular logic of OCN connectivity.

OCNs fall into two major groups: medial olivocochlear neurons (MOCs), which sit in the ventral nucleus of the trapezoid body (VNTB), and lateral olivocochlear neurons (LOCs), which are intermingled with afferent principal neurons in the lateral superior olive (LSO; *Figure 1A*). Both MOCs and LOCs extend peripheral axons that spiral along the length of the cochlea, often spanning large frequency domains (*Brown, 2011*; *Figure 1B*). MOCs terminate mainly on outer hair cells (OHCs), a group of mechanosensory cells that influence cochlear responses to auditory stimuli through rapid, sound-induced cell-body movements (*van der Heijden and Vavakou, 2022*; *Ashmore, 2019*). It is widely accepted that MOCs affect cochlear gain through inhibitory, cholinergic synapses that reduce OHC motility (*Mountain, 1980*; *Siegel and Kim, 1982*; *Blanchet et al., 1996*; *Dallos et al., 1997*; *Fuchs and Lauer, 2019*), much as visual efferent pathways control gain by altering pupil diameter. As

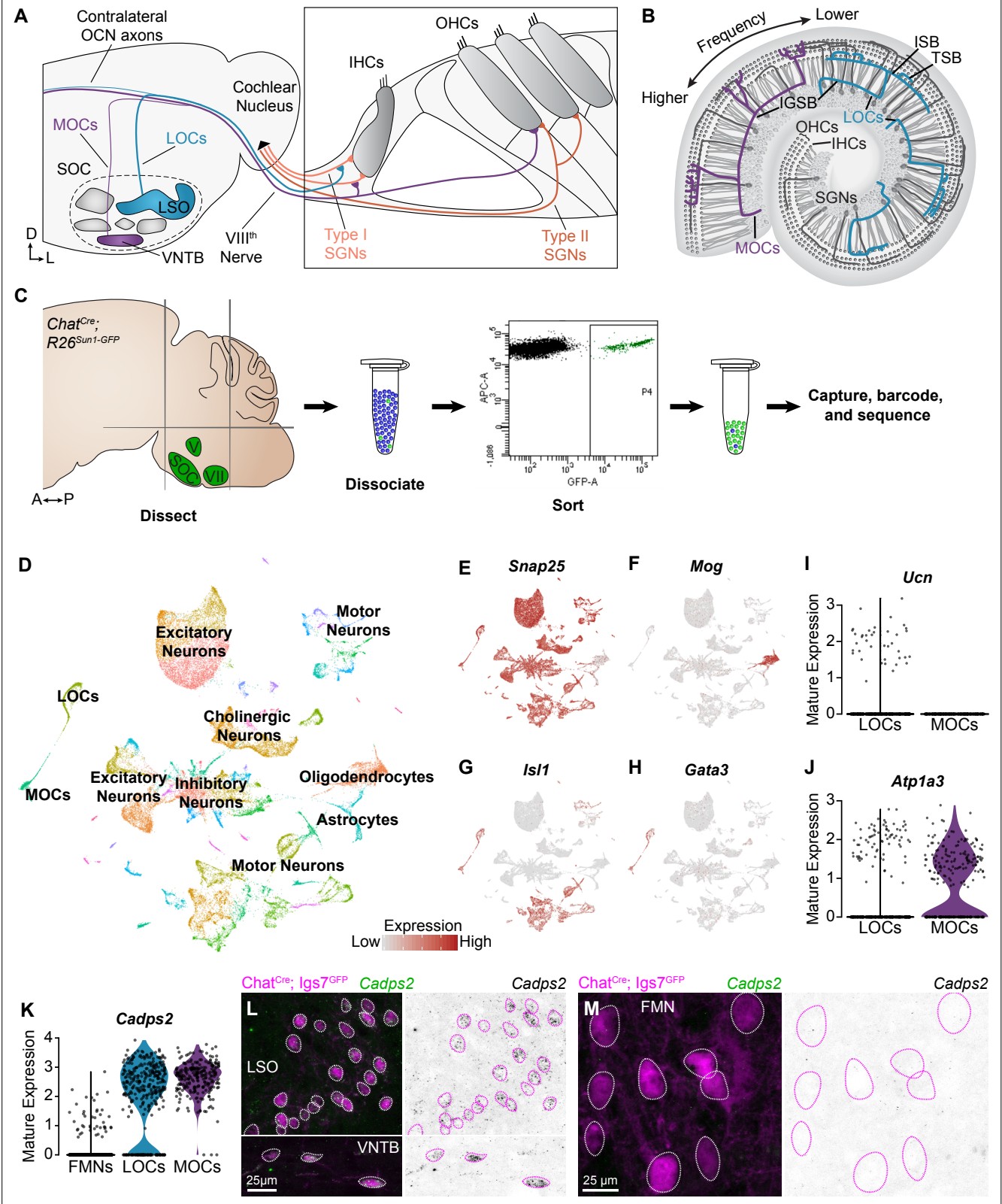

**Figure 1.** Identification of OCNs by single-nucleus sequencing of brainstem neurons. (**A, B**) Schematic of medial (MOC) and lateral (LOC) olivocochlear neuron (OCN) subtypes housed in the brainstem (**A**) and their projections into the cochlea, shown in cross-section (**A**) and top-down (**B**). D, dorsal; L, lateral; LSO, lateral superior olive; VNTB, ventral nucleus of the trapezoid body; SOC, superior olivary complex; IHC, inner hair cell; OHC, outer hair cell; SGN, spiral ganglion neuron; IGSB, intraganglionic spiral bundle; ISB, inner spiral bundle; TSB, tunnel spiral bundle. (**C**) Schematic of single-nucleus

*Figure 1 continued on next page*

*Figure 1 continued*

collection protocol. The ventral brainstem of *Chat^Cre^; Rosa26^Sun1-GFP^* mice was dissected, including the SOC, facial motor nucleus (VII), and trigeminal motor nucleus (V), followed by collection and sequencing of individual GFP+ nuclei. A, anterior; P, posterior. (**D**) UMAP plot summarizing sequencing data of 45,828 single nuclei from 61 animals across three ages. The data includes 56 clusters, including MOCs, LOCs, and several different motor neuron types, as well as a few non-neuronal populations. (**E–H**) Feature plots denoting normalized expression levels of selected genetic markers. For visualization purposes, high expression on the color scale is capped at the 95^th^ percentile of gene expression. The majority of clusters express the neuronal marker *Snap25* (**E**). The main non-neuronal population is labeled by *Mog*, an oligodendrocyte marker (**F**). Motor neuron clusters express the motor neuron determinant *Isl1* (**G**). Among these brainstem motor neurons, OCNs are identified by expression of the transcription factor *Gata3* (**H**). (**I, J**) Differential expression of known markers *Ucn* (**I**) and *Atp1a3* (**J**) denotes LOCs and MOCs, respectively. (**K–M**) *Cadps2* is a novel marker for OCNs that is expressed by both MOCs and LOCs but not FMNs. (**K**) Violin plots summarizing transcriptional expression of *Cadps2* in mature LOCs, MOCs, and FMNs. (**L, M**) Fluorescent *in situ* hybridization (FISH) verifies that *Cadps2* is detected in both MOCs and LOCs (**L**) but not in FMNs (**M**). Representative images are shown from a total of three animals of either sex, P28.

The online version of this article includes the following figure supplement(s) for figure 1:

**Figure supplement 1.** Quality-control metrics for single-nucleus sequencing data.

**Figure supplement 2.** OCNs and FMNs are transcriptionally distinct.

well as protecting cochlear circuitry from the impact of acoustic trauma (*Reiter and Liberman, 1995*; *Taranda et al., 2009*; *Maison et al., 2013*; *Boero et al., 2018*; *Fuente, 2015*; *Rajan, 1995*; *Tong et al., 2013*), MOCs are thought to contribute to the detection of signals in noise (*Guinan, 2018*; *Kawase et al., 1993*; *Winslow and Sachs, 1987*) and to tune sensory responses based on states of attention (*Delano et al., 2007*; *Terreros et al., 2016*; *Oatman, 1976*; *Oatman, 1971*). Consistent with this range of effects on the perception of sound, electrophysiological characterization suggests that MOCs are controlled both by inputs from the cochlear nucleus and by descending inputs from multiple sources (*Romero and Trussell, 2022*).

In contrast, we know little about the LOCs, as they are difficult to access for recording, stimulation, or surgical ablation. Anatomically, LOCs terminate on the peripheral processes of Type I spiral ganglion neurons (SGNs), primary sensory neurons that carry perceptual auditory information from mechanosensory inner hair cells (IHCs) into the brain. Functionally, the nature of communication between LOCs and SGNs remains opaque: although LOCs express acetylcholine, a variety of other signaling molecules have also been reported in these cells, including GABA, CGRP, dopamine, enkephalin, dynorphin, and urocortin (Ucn) (*Eybalin, 1993*; *Kitcher et al., 2022*; *Sewell, 2011*; *Safieddine and Eybalin, 1992*; *Safieddine et al., 1997*; *Kaiser et al., 2011*; *Takeda et al., 1987*; *Adams et al., 1987*). However, definitive evidence for many of these transmitters is lacking. Moreover, indirect activation of the auditory efferent system can elicit both excitatory and inhibitory effects on SGNs, hinting at the possibility of multiple LOC subtypes that can direct distinct effects on their downstream targets (*Groff and Liberman, 2003*). No specific role in hearing has been definitively linked to LOCs, although they have been implicated in sound localization (*Darrow et al., 2006a*; *Larsen and Liberman, 2010*; *Irving et al., 2011*) and appear to protect the ear from noise damage (*Fuente, 2015*; *Darrow et al., 2007*; *Kujawa and Liberman, 1997*). The mechanisms underlying this protective effect remain unclear. One promising model holds that dopamine—and potentially other neurotransmitters—released from LOCs inhibits SGN firing, thereby dampening excitotoxicity (*Wu et al., 2020*; *Ruel et al., 2001*). However, this model has yet to be tested directly, and other, as yet unidentified, pathways may contribute a protective role as well.

Here, we took a multi-pronged approach to investigate key molecular, anatomical, and physiological features of the auditory efferent system. We found many novel transcriptional features of MOCs and LOCs in both developing and mature OCNs, including cell-type specific markers, genes that could confer distinct physiological properties, and developmental gene-expression changes that highlight pathways involved in OCN maturation. In addition, we found that LOCs cluster into two molecularly and anatomically distinct subtypes based on expression of neuropeptides. However, regardless of the amount of neuropeptide Y in their pre-synaptic puncta, LOCs elaborate axon arborizations that vary extensively and terminate on multiple SGN subtypes. Further, LOCs change their neuropeptide expression profiles upon the onset of hearing and after acoustic trauma, indicating that variability in peptide expression may serve as a way to modulate sensory circuits based on prior experience.

# Results

To assess transcriptional variability within OCNs, we sequenced cell nuclei from individual cholinergic neurons, which were labeled using *Chat^Cre* to drive expression of a nuclear-localized GFP (*Rosa26^Sun1-GFP*) (*Rossi et al., 2011*; *Mo et al., 2015*). We enriched for OCNs and the closely related FMNs by dissecting a region of the ventral brainstem that includes the SOC as well as facial and trigeminal motor neurons. We then used fluorescence-activated cell-sorting (FACS) to isolate dissociated GFP+ cholinergic nuclei and employed the 10x Genomics platform to encapsulate nuclei and generate barcoded single-nucleus libraries (*Figure 1C*). To examine the maturation of these cell types and identify markers that are expressed consistently across postnatal development, nuclei were collected at two pre-hearing timepoints, postnatal day (P)1 (n=13 animals) and P5 (n=16 animals), the latter of which is an important time of synapse refinement in auditory circuits (*Frank and Goodrich, 2018*; *Yu and Goodrich, 2014*). We also collected nuclei at P26–P28 (n=32 animals), when auditory circuitry is grossly mature. After filtering out low-quality cells and infrequently expressed genes (see Materials and methods), our dataset includes 45,828 nuclei: 16,753 cells from P1 animals; 15,542 from P5 animals; and 13,533 cells from P26–P28 animals. Unsupervised, graph-based clustering analysis identified 56 clusters (*Figure 1D*). These clusters primarily consist of neurons, as indicated by the expression of neuronal markers like *Snap25* (*Figure 1E*). Populations of oligodendrocyte precursors, *Mog*-positive oligodendrocytes (*Figure 1F*), and astrocytes were also identified based on expression of well-established markers (*Zhang, 2001*). Each neuronal cluster includes cells from all three timepoints (*Figure 1—figure supplement 1A*). Non-neuronal cells primarily originate from the adult dataset, consistent with prior reports showing an increase in myelination of auditory brainstem neurons and elevated expression of oligodendrocyte markers over the first few postnatal weeks (*Long et al., 2018*). Although cell types vary somewhat in the number of genes and unique molecular identifiers (UMI) detected, the structure of the data is not driven by these technical variables (*Figure 1—figure supplement 1B–D*). Cell types were also similar in the fraction of genes mapping to the mitochondrial genome, suggesting that clusters are not affected by differences in cell health (*Ilicic et al., 2016*; *Luecken and Theis, 2019*; *Figure 1—figure supplement 1E*). The neuronal clusters include several motor neuron clusters, as defined by co-expression of *Isl1* (*Figure 1G*), *Tbx20*, and *Phox2b*. Among these, two OCN clusters were identified based on expression of *Gata3*, which is expressed in OCNs but not in canonical motor neuron populations (*Figure 1H*; *Karis et al., 2001*). LOC and MOC clusters were identified by expression of the LOC-specific peptide Urocortin (*Ucn*) (*Kaiser et al., 2011*; *Figure 1I*) and the Na,K-ATPase *Atp1a3*, which is expressed in MOCs but not LOCs in mature rats (*McLean et al., 2009*; *Figure 1J*).

## Gene-expression profiles that distinguish LOCs and MOCs from FMNs and each other

LOCs and MOCs arise from a common progenitor pool with and develop alongside FMNs, raising the question of how OCNs deviate from a typical motor neuron fate to play such an unorthodox role in peripheral sensory modulation (*Frank and Goodrich, 2018*; *Fritzsch and Elliott, 2017*). To learn more about the transcriptional origins of their unique developmental and mature properties, we compared all OCNs to a cluster of FMNs, which was identified based on co-expression of *Etv1* and *Epha7* (*Figure 1—figure supplement 2A, B*; *Tenney et al., 2019*). Our analysis revealed substantial transcriptional differences between OCNs and FMNs that could contribute to their distinct functions and anatomy (*Figure 1—figure supplement 2C*). For instance, OCNs show selective expression of *Cadps2*, which encodes the calcium-dependent activator protein for secretion 2 (CADPS2, also known as CAPS-2; *Figure 1K*). CADPS2 is a member of the Munc13 and CAPS family of priming proteins (*Imig et al., 2014*; *Jockusch et al., 2007*; *Nestvogel et al., 2020*) and has been linked to exocytosis of dense-core vesicles (*Tandon et al., 1998*; *Speidel et al., 2003*; *Grishanin et al., 2004*; *Renden et al., 2001*), consistent with neuropeptide release from OCNs (*Eybalin, 1993*; *Kitcher et al., 2022*). Fluorescent *in situ* hybridization (FISH) in P27–P28 mice confirmed that *Cadps2* is expressed in both MOCs and LOCs but is not detectable in FMNs (*Figure 1L and M*; n=3 animals). This finding validates our OCN and FMN clusters and establishes a new OCN-specific marker.

Despite their early shared origins, FMNs and OCNs express unique combinations of genes encoding transcription factors, adhesion molecules, and other receptors needed for proper neuronal migration, targeting, and synaptic specificity. These differences are particularly prominent at P1 and

P5, while OCN axons are still growing to their final targets in the inner ear and before many developmentally salient molecules become downregulated (*Figure 1—figure supplement 2D, E*). Cell-type-specific transcription factor genes include *Pbx3* in FMNs and *Sall3* in OCNs (*Figure 1—figure supplement 2G*). Genes for many guidance and adhesion molecules are also differentially expressed, including *Kirrel3* in OCNs and *Pcdh15* in FMNs (*Figure 1—figure supplement 2H*). Consistent with their integration into fundamentally different circuitry, OCNs and FMNs also differ in the expression of molecules relevant to mature function (*Figure 1—figure supplement 2F, I*). In particular, OCNs express *Gad2*, a gene involved in GABA synthesis, whereas FMNs express the serotonin receptor gene *Htr2c* (*Figure 1—figure supplement 2I*).

Although OCNs share several properties that broadly distinguish them from FMNs, OCNs are themselves heterogeneous, with LOCs and MOCs playing distinct roles in auditory circuitry (*Guinan, 2018*). To date, however, few reliable markers for either population have been identified. For instance, although *Atp1a3* is enriched in MOCs in adulthood, it is expressed in both LOCs and MOCs developmentally (*Figure 2—figure supplement 1B, C*). To address this gap, we identified several genes that are enriched in either the MOC or LOC cluster across all three timepoints and verified their expression in mature animals by FISH (*Figure 2A–G*, *Figure 2—figure supplement 1A–G*). We found that *Col4a4* is expressed in virtually all LOCs in mature (P26–P28) animals, whereas *Zfp804a* is expressed in virtually all MOCs (*Figure 2B–G*). These findings further validate the identity of MOC and LOC clusters and present new markers to distinguish MOC and LOC neurons across development.

In mature animals, LOCs and MOCs are integrated into distinct circuits, receiving input from multiple brain regions and responding to numerous neurotransmitters and modulatory substances (*Brown, 2011*; *Romero and Trussell, 2022*). Consistently, our data point to many subtype-specific transcription factors and adhesion molecules that could direct their unique developmental trajectories, including migration to different nuclei in the SOC and guidance of axons to different target cells in the cochlea (*Figure 2H, I*). In addition, genes for many physiologically relevant receptors and ion channels are differentially expressed (*Figure 2J*), including receptors and receptor subunits for neurotransmitters such as acetylcholine, glutamate, and orexin (*Figure 2K–R*). Differential expression analysis between neonatal and mature timepoints identified numerous genes, including several that encode neurotransmitter receptors and ion channels (*Figure 2S*, *Figure 2—figure supplement 1H, I*), supporting a recent report that mature physiological properties of LOCs emerge gradually (*Hong et al., 2022*). We also identified altered expression of transcription factor genes, like *Auts2*, that may orchestrate these developmental changes (*Figure 2S*).

## A novel, anatomically segregated subtype of LOCs is defined by neuropeptide expression

One challenge in understanding LOC function has been the presence of additional heterogeneity within this population (*Brown, 2011*; *Romero and Trussell, 2022*). Therefore, we sub-clustered mature OCNs, focusing on differences among LOCs. This analysis revealed five OCN clusters (*Figure 3A*). Quality-control metrics—including the number of genes detected and originating batch—were similar among all clusters (*Figure 3—figure supplement 1A–D*). Based on expression of MOC and LOC markers, we identified a large cluster of MOCs (230 cells) (*Figure 3B*) and two main clusters of LOCs (242 LOC1 cells, 75 LOC2 cells; *Figure 3C*). This analysis also revealed two smaller clusters with only 21 cells each. One of these clusters (MOC2) expressed MOC markers, including *Zfp804a*; the other appears to consist of miscellaneous cells (Misc), as these cells did not express either MOC or LOC markers. We identified only a small number of differentially expressed genes in the MOC2 and Misc clusters (*Figure 3—figure supplement 1E, F*), possibly due to low cell counts. We were therefore unable to definitively identify either cluster and did not pursue them further.

Comparison of the two LOC subtypes revealed that LOC2s are distinguished by upregulation of genes for the neuropeptides CGRP (*Calca*), CGRP-II (*Calcb*), NPY, and Ucn (*Figure 3D and J-M*). LOC1s and LOC2s also differ in expression of the genes for the transcription factor Meis1 and for guidance and adhesion molecules, such as Tenm2, Tenm3, Unc5c, and Epha6 (*Figure 3E–I*), suggesting that these clusters indeed represent distinct cell types rather than reflecting the same pool of cells in different states. Consistent with this idea, LOC2s are confined to the medial wing of the LSO (n=4 animals of either sex, P28–P32), as revealed by immunostaining for CGRP and Ucn in tissue from an *Npy-GFP* reporter mouse (*Figure 3O*). Peptide expression is heterogeneous: although all adult

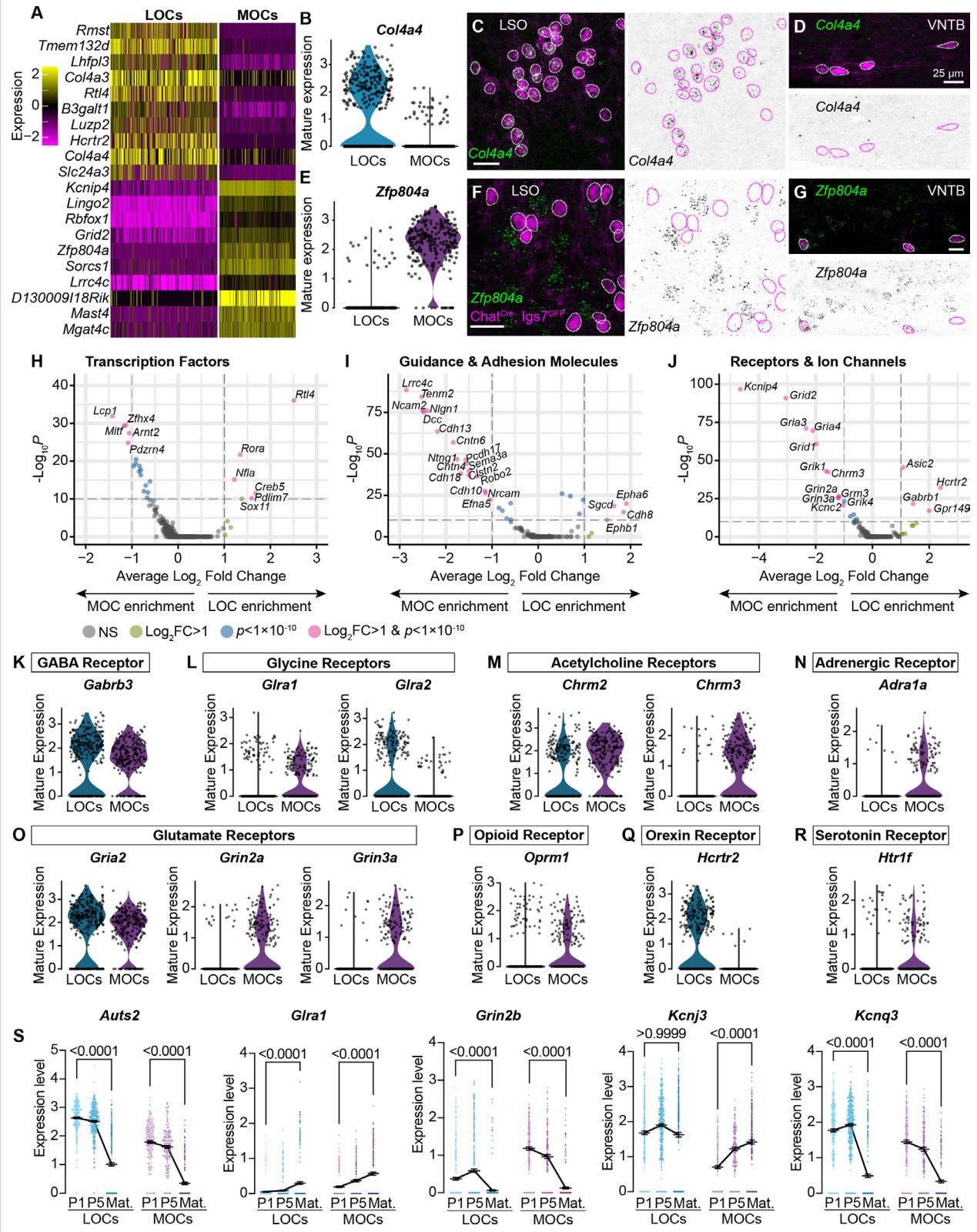

**Figure 2.** MOCs and LOCs exhibit distinct gene-expression profiles. (**A**) Heatmap summarizing expression of the 10 genes with the highest average log-fold change for LOCs and MOCs. For visualization purposes, scaled expression levels were capped at –2.5 and 2.5. (**B–G**) Fluorescent *in situ* hybridization in *Chat^Cre^; Igs7^GFP^* animals (**C, D, F, G**) validates novel markers with enriched expression in sequencing data from LOCs or MOCs, shown in violin plots (**B, E**). The LOC-enriched gene *Col4a4* (green) is expressed by GFP+, cholinergic neurons (magenta) in the LSO (**C**) but not in the VNTB

*Figure 2 continued on next page*

*Figure 2 continued*

(**D**), shown also as inverted images with magenta circles around each cholinergic neuron. In contrast, the MOC-enriched gene *Zfp804a* is expressed in MOCs (**G**) but is absent from LOCs (**F**). Representative images are shown from three animals of either sex, P28. (**H–J**) Volcano plots indicating differentially expressed genes between mature MOCs and LOCs that encode transcription factors (**H**), guidance and adhesion molecules (**I**), and receptors and ion channels (**J**). Gene names are listed for all genes with an average log$_2$-fold change greater than 1 and adjusted p<1 × 10$^{-10}$ (dashed lines; Wilcoxon rank-sum test, Bonferroni post-hoc correction). (**K–P**) Violin plots denoting normalized expression levels of genes for neurotransmitter receptor subunits. Both MOCs and LOCs generate transcripts corresponding to receptors for numerous neurotransmitters, including GABA (**K**), Glycine (**L**), acetylcholine (**M**), and glutamate (**O**). In several cases, however, MOCs and LOCs express genes for different subunits or classes of receptors (**L, M, O**). MOCs and LOCs also differ in their expression of genes that encode more specialized receptors, including the adrenergic receptor *Adra1* (**N**), mu opioid receptor *Oprm1* (**P**), orexin receptor *Hcrtr2* (**Q**), and serotonin receptor *Htr1f* (**R**). (**S**) Log-normalized counts of transcripts for the autism-related transcription factor *Auts2* and several neurotransmitter receptor and ion channel genes whose expression changes across postnatal development (*Glra1*, *Grin2b*, *Kcnj3*, and *Kcnq3*). Kruskal-Wallace with Dunn's test for multiple comparisons. Error bars, mean ± SEM.

The online version of this article includes the following figure supplement(s) for figure 2:

**Figure supplement 1.** OCN markers across postnatal development.

LOCs express some level of CGRP, as previously reported (*Brown, 2011*; *Wu et al., 2018*; *Maison et al., 2003a*), LOC2s did not uniformly express both NPY and Ucn (*Figure 3N*; *Supplementary file 1*). NPY seems to be a particularly reliable marker for LOC2s, as Ucn was not always expressed in NPY+ cells and very few cells expressed only Ucn. Furthermore, an intersectional genetic approach in which *Npy*$^{FlpO}$; *Chat*$^{Cre}$ drives expression of the dual recombinase *Rosa26*$^{FLTG}$ reporter recapitulates the pattern of NPY+ LOC restriction to the medial wing of the LSO (*Figure 3P*), without labeling MOCs in the VNTB (arrowheads, *Figure 3P*) or the lateral wing of the LSO (n=4 animals of either sex, P28–P30). Thus, we propose that LOC2s comprise a distinct cell type that is poised to release multiple neuropeptides in the cochlea.

Previous studies have indicated that LOCs produce an array of signaling molecules, including acetylcholine, GABA, CGRP, and opioids (*Eybalin, 1993*; *Safieddine and Eybalin, 1992*). Having confirmed expression of *Chat*, *Gad2*, and *Calca* across OCNs (*Figure 1—figure supplement 2F*, *Figure 3—figure supplement 1G*), we sought to verify whether transcripts for any opioid precursors were detected in either LOC subtype. However, we were unable to observe expression of either *Penk* or *Pdyn* in any LOC subtype in either our sequencing data or via FISH (*Figure 3—figure supplement 1G, H*). The lack of transcripts implies that neither LOC subtype produces opioid peptides, although we cannot rule out the presence of a minority population or that expression is transient.

## Anatomically diverse OCNs broadly innervate the cochlea

Anatomical studies suggest that the frequency distribution of LOC axons is organized in a stereotyped manner, with medial neurons projecting to higher-frequency regions of the cochlea and lateral neurons projecting to lower-frequency regions (*Brown, 2011*; *Robertson et al., 1987*; *Guinan et al., 1984*). These projections are consistent with the tonotopic organization of LSO principal neurons, which are arranged from low-to-high frequencies along a lateral-to-medial axis (*Kandler et al., 2009*). The confinement of LOC2s to the medial LSO therefore raises the possibility of tonotopically restricted effects in the cochlea. However, although the density of presumptive LOC2 axons, labeled by *Npy*$^{FlpO}$; *Chat*$^{Cre}$; *Rosa26*$^{FLTG}$, decreases from high-frequency (basal) to low-frequency (apical) regions, NPY+ LOC axons span the entirety of the base-to-apex length (*Figure 3Q*, left to right).

Because OCN axons can arborize extensively along the length of the cochlea, the broad distribution of LOC2 processes observed in *Npy*$^{FlpO}$; *Chat*$^{Cre}$; *Rosa26*$^{FLTG}$ cochleae might obscure meaningful differences in synaptic connectivity. Indeed, a wide variety of LOC morphologies has been described (*Brown, 2011*; *Warr et al., 1997*; *Brown, 1987*), raising the possibility that LOC2s correspond to one anatomically distinct group. To analyze the morphology and synaptic connectivity of individual OCN axons, we leveraged our discovery that *Ret* is expressed by all OCNs from early in development (*Figure 5—figure supplement 1A, B*) and used low doses of tamoxifen in *Ret*$^{CreER}$ mice to drive expression of the Cre-dependent *Igs7*$^{GFP}$ reporter in a small number of OCNs (*Figure 5—figure supplement 1C, D*; *Luo et al., 2009*). Labeled OCNs colocalize with ChAT immunolabeling in the brainstem (*Figure 5—figure supplement 1C, D*), and a subset of LOCs are also positive for NPY (*Figure 5—figure supplement 1C'*, closed arrowhead), confirming that this approach captures both LOC1s and LOC2s. Sparsely labeled OCN axons display peripheral innervation patterns consistent

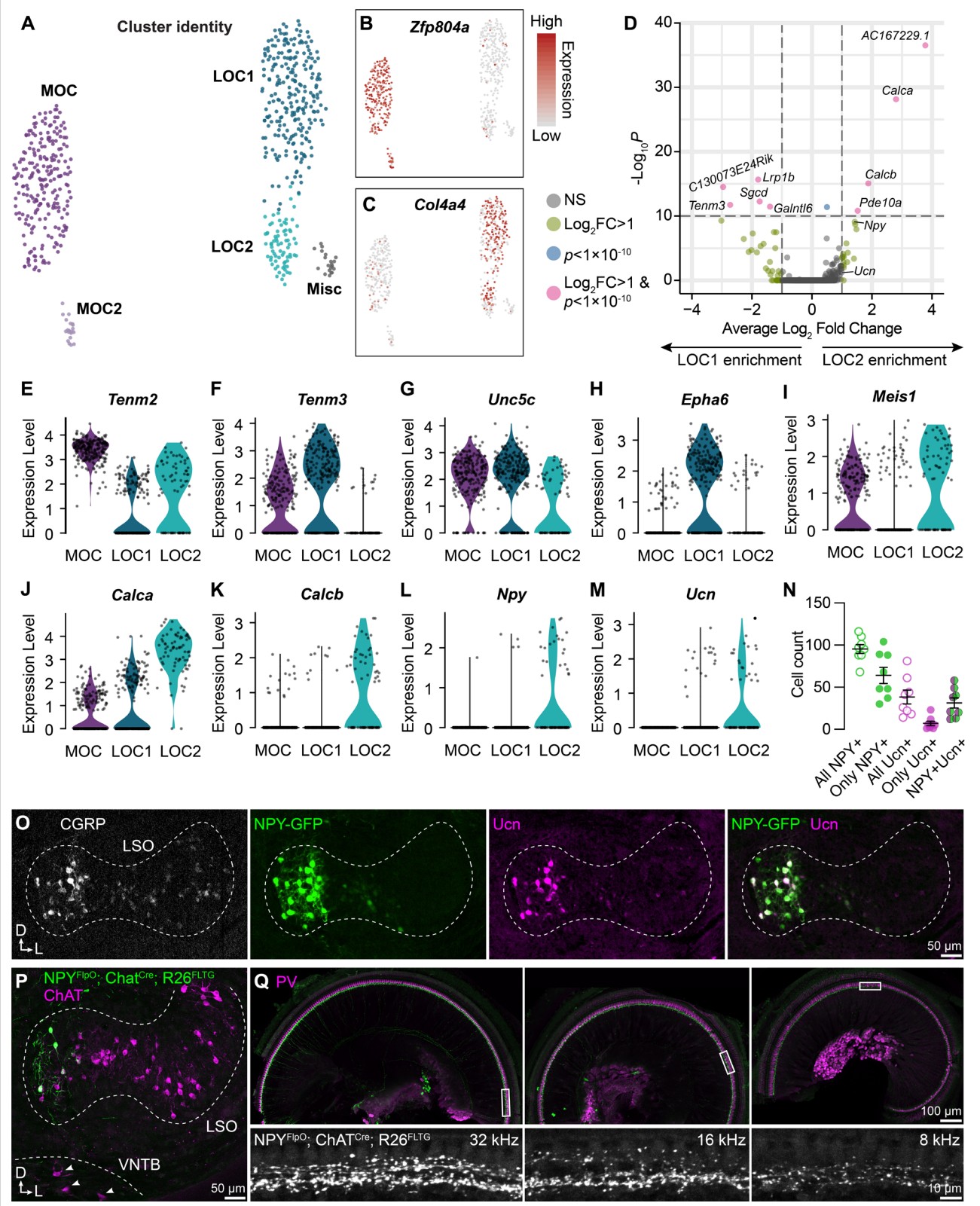

**Figure 3.** LOC neurons include a peptide-enriched subset. (**A–C**) UMAPs summarizing sub-clustering results of 590 OCNs from 32 *Chat^Cre^; Rosa26^Sun1-GFP^* animals (P26–P28). OCNs sub-cluster into five different groups (**A**). Feature plots denote normalized expression levels of the MOC marker *Zfp804a* (**B**) and the LOC marker *Col4a4* (**C**). For visualization, the upper level of expression was capped at the 95th percentile. (**D**) Volcano plot summarizing differences in gene expression between LOC1 and LOC2 clusters. Wilcoxon rank-sum test, Bonferroni post-hoc correction. (**E–M**) Violin plots illustrating

*Figure 3 continued on next page*

*Figure 3 continued*

expression of genes encoding adhesion molecules (**E–H**), transcription factors (**I**), and neuropeptides (**J–M**) that differ among OCN subtypes. (**N, O**) Representative images (**O**) and quantification (**N**) of CGRP and Ucn antibody stains on *Npy-GFP* tissue showing biased expression in the medial arm of the LSO. n=4 animals of either sex, P28–P32. Error bars, mean ± SEM. (**P**) Representative image of ChAT antibody stain on *Npy^FlpO^; Chat^Cre^; Rosa26^FLTG^* tissue illustrating intersectional genetic labeling of NPY-enriched LOCs in the medial arm of the LSO. MOCs in the VNTB (arrowheads) are never labeled in this mouse line. n=4 animals of either sex, P28–P32. (**Q**) Top-down view of a representative *Chat^Cre^; Npy^FlpO^; Rosa26^FLTG^* cochlea (left to right: base, mid, apex) that was stained for parvalbumin (PV, magenta) to label HCs and SGNs. Insets below each panel show genetically labeled NPY+ LOCs in the ISB beneath the IHCs at three different frequency regions. n=10 animals of either sex, P28–P32.

The online version of this article includes the following figure supplement(s) for figure 3:

**Figure supplement 1.** Additional metrics for OCN transcriptional subtypes.

with what has been described (*Brown, 2011*; *Warr et al., 1997*; *Brown, 1987*; *Figure 5—figure supplement 1E–E"*). To better define the connectivity of individual *Ret^CreER^; Igs7^GFP^*-labeled MOC and LOC axons, cochleae were also stained for the pre-synaptic marker synaptophysin (Syp) to mark putative OCN synapses and calbindin-2 (CALB2, also known as calretinin) to label IHCs and Type I SGN subtypes (*Shrestha et al., 2018*; *Sun et al., 2018*; *Petitpré et al., 2018*; *Figure 4A*). Putative pre-synaptic sites in each axon were isolated using the fluorescence signal from labeled OCN axons to mask the Syp signal (*Figure 4A'*, *Figure 4—video 1*, *Figure 5—video 1*).

Analysis of MOC innervation patterns confirmed that this sparse labeling approach recapitulates known morphologies and synaptic distributions. Sparsely labeled MOC axons (n=40 axons, N=9 animals of both sexes) have terminal morphologies consistent with prior dye-labeling studies (*Brown, 2011*; *Brown, 1987*; *Robertson and Gummer, 1985*; *Brown, 2014*), including relatively simple axons with little-to-no branching beneath the OHCs (*Figure 4A" and B*); branching in the OSL (*Figure 4C*); complex branching beneath IHCs and among OHCs (*Figure 4D*) or branching in a goal-post fashion (*Figure 4E*). Labeled MOCs show a slight preference for innervating the third row of OHCs, with 21.4% of Syp puncta in row 1, 31.6% in row 2, and 47% in row 3. Additionally, 45% of the reconstructed MOC axons (18/40) have Syp puncta in the inner spiral bundle (ISB), localized among SGN peripheral fibers, either *en passant* (*Figure 4A"*) or, less frequently, at terminal endings (*Figure 4F*). This observation confirms previous reports that a subset of MOC terminal axons innervate SGN processes, although the number of putative pre-synaptic sites within a given MOC axon are somewhat lower than what was identified in EM reconstructions (*Robertson and Gummer, 1985*; *Hua et al., 2021*). This suggests that our approach is reliable and, if anything, may undercount pre-synaptic sites.

Because the tamoxifen induction approach labeled multiple OCNs in a given animal, we could not reliably associate individual terminal axons to a particular labeled soma in the brainstem. Instead, our analyses consider each terminal axon independently. However, in one animal, a single MOC soma was labeled in the brainstem. We were therefore able to confidently associate multiple terminal axons in the corresponding cochlea to the same MOC cell body (*Figure 5—figure supplement 1E–E"*). Six terminal fibers stemming from this single MOC innervated discrete patches along a~27 kHz swath of the cochlea (*Figure 4A"*, *Figure 5—figure supplement 1F–I*). Within this one MOC, some (shown in *Figure 4A–A"*), but not all (*Figure 5—figure supplement 1F–I*), of its terminal fibers contained Syp puncta in the ISB. This suggests that there is not a discrete morphological class of MOCs that makes branches that consistently contact both SGN peripheral fibers and OHCs. Likewise, when considering the entire dataset, there were no significant differences in the frequency of innervation (*Figure 4G*) or number of branch points (*Figure 4H*) between terminal MOC axons that did and did not contain Syp puncta in the ISB. Collectively, these observations indicate that individual MOCs can produce projections with variable morphologies and synaptic targets.

## NPY-positive LOC axons exhibit highly variable patterns of connectivity

We next asked whether LOC2s, as identified by NPY expression, exhibit any stereotyped axon morphologies or patterns of connectivity with different SGN subtypes in the cochlea. There are three physiologically and molecularly distinct Type I SGN subtypes whose terminals are arranged along the base of IHCs in a stereotyped manner (*Figure 5A*; *Shrestha et al., 2018*; *Sun et al., 2018*; *Petitpré et al., 2018*; *Liberman, 1978*; *Liberman et al., 2011*; *Liberman, 1982*; *Petitpré et al., 2020*). This diversity is thought to enable the encoding of a wide range of sound intensities (*Petitpré et al., 2020*). Type Ia SGNs express high levels of CALB2 and form synapses on the pillar side of IHCs, nearer

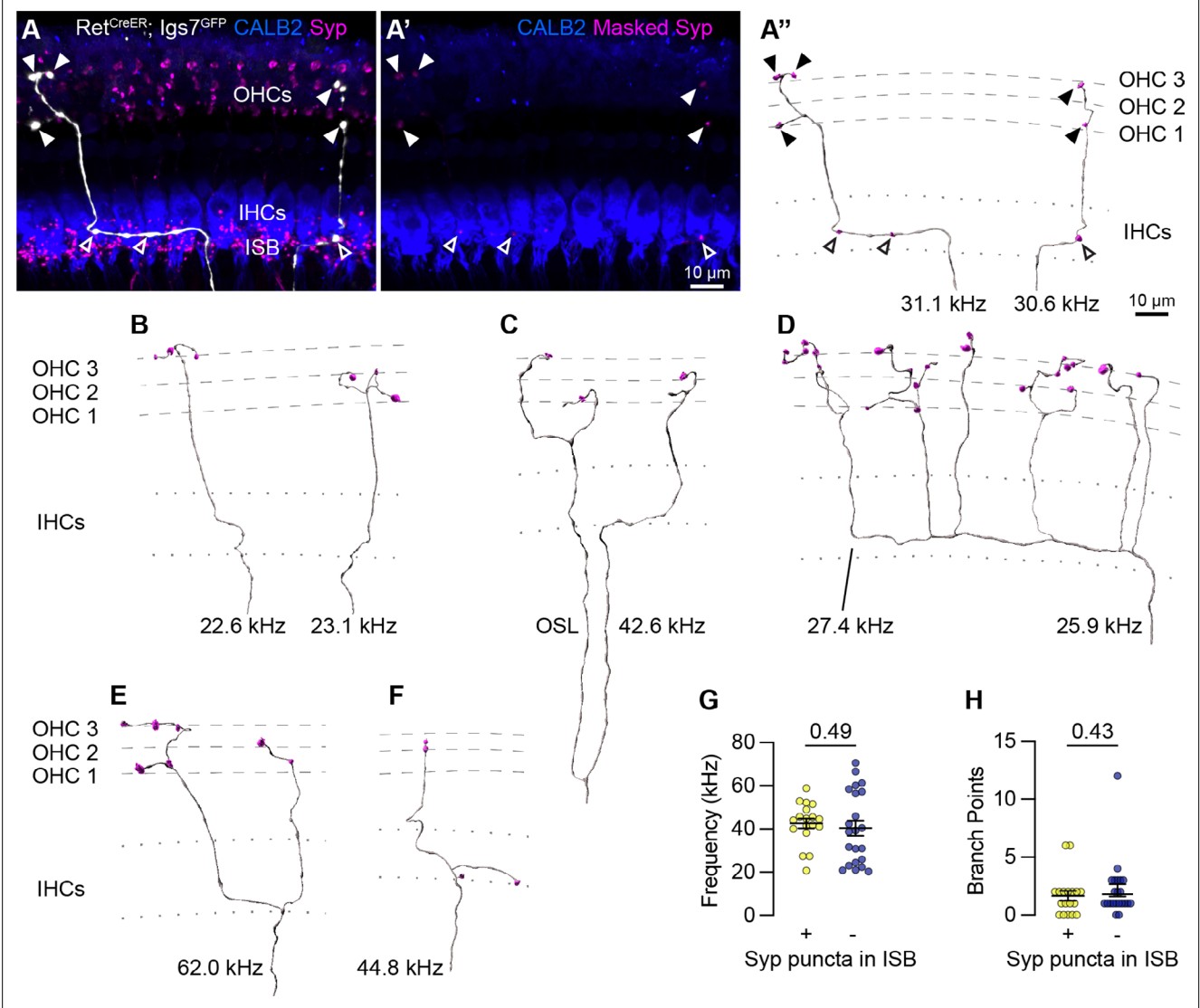

**Figure 4.** Sparse labeling of OCNs reveals MOC connectivity in the cochlea. (**A–A''**) Low doses of tamoxifen in *Ret^CreER*; *Igs7^GFP* animals sparsely labels OCNs, enabling visualization of individual MOC axons (white) as they extend along and through the row of IHCs (CALB2, blue), and form terminal swellings among the OHCs, with Syp (magenta) labeling putative pre-synaptic sites (**A**). Using the GFP signal to mask the Syp fluorescent signal isolates Syp puncta associated with the labeled axon (**A'**), shown in an Imaris reconstruction in (**A''**). Closed arrowheads, Syp puncta in OHC region; open arrowheads, Syp puncta beneath IHCs. Dashed lines indicate OHC rows; dotted lines indicate the top and bottom of IHCs. Additional reconstructions from this same MOC neuron are shown in *Figure 5—figure supplement 1F-I*. (**B–F**) Terminal MOC axons exhibit a range of morphologies, including simple and unbranched (**B**), branching in the OSL and among the OHCs (**C**), highly branched in both the ISB and OHC region (**D**), and branching in the ISB with simple collaterals innervating multiple rows of OHCs (**E**). Some MOC axons also have Syp puncta in the ISB, either along an axon (**A''**) or on a terminal branch (**F**). The frequency of each axon is indicated (kHz); base is to the left. (**G–H**) MOCs with and without Syp puncta in the ISB do not differ in their frequency (**G**) or number of branch points (**H**). Error bars, mean ± SEM. Wilcoxon rank-sum test. All reconstructions are from P27–P30 cochleae. n=40 axons, N=9 animals of both sexes.

The online version of this article includes the following video for figure 4:

**Figure 4—video 1.** Terminal MOC axon morphology with putative synapses in the ISB and OHC region.

https://elifesciences.org/articles/83855/figures#fig4video1

the Tunnel of Corti and OHCs (*Figure 5A*, dark blue). On the other hand, Type Ic SGNs express low levels of CALB2 and synapse on the modiolar side of IHCs, nearer the SGN cell bodies (*Figure 5A*, light blue). Type Ib SGNs are defined by moderate levels of CALB2, likely filling in the space between the Ia and Ic subtypes along the base of the IHC. In *Npy^FlpO*; *Chat^Cre*; *Rosa26^FLTG* cochlea, genetically labeled LOC2 axons intermingle closely with parvalbumin (PV)+ SGN peripheral processes on both

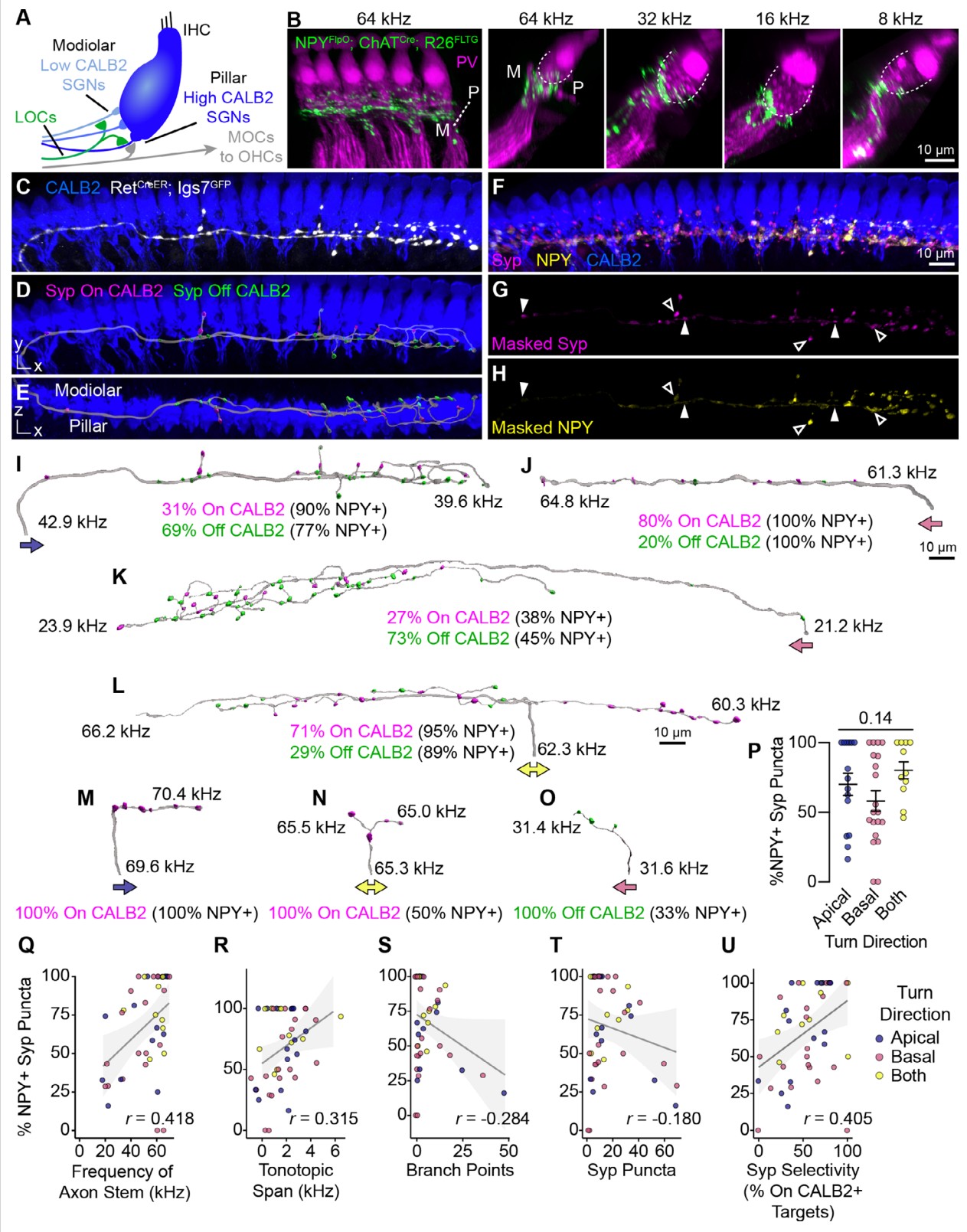

**Figure 5.** Anatomical heterogeneity in LOCs does not correlate with peptidergic identity. (**A**) Schematic of the IHC region of the Organ of Corti. Peripheral fibers of Type I SGN subtypes (blue) innervate the IHCs in a low-to-high CALB2 expression gradient along the modiolar-pillar axis. (**B**) Intersectional genetic approach using *Npy^FlpO*; *Chat^Cre*; *Rosa26^FLTG* labels peptidergic LOC axons in the cochlea (green). Parvalbumin (PV, magenta) labels IHCs and SGN peripheral processes. GFP-positive peptidergic LOC axons are spread across much of the modiolar-pillar (M–P) axis in the ISB beneath

*Figure 5 continued on next page*

*Figure 5 continued*

IHCs, shown in orthogonal views of 1–2 IHCs at frequencies from 64 to 8 kHz. (**C–I**) 3D views of an individual LOC axon in a sparsely labeled *Ret^CreER^*; *Igs7^GFP^* cochlea. Fluorescent signal from a genetically labeled LOC (**C**, white) was used to reconstruct individual terminal axons in Imaris, shown in the same perspective (**D, I**) and an orthogonal view looking from the top of the IHCs to the bottom (**E**). (**F**) Cochlear wholemounts were immunolabeled for CALB2 (blue), synaptophysin (Syp, magenta), and NPY (yellow). Using the fluorescent signal from the LOC, masks were created for Syp (**G**) and NPY (**H**), allowing for identification of NPY+ and NPY- (open and closed arrowheads, respectively) Syp puncta associated with an individual axon. Using a surface reconstruction of the CALB2 immunofluorescence signal, Syp puncta were also categorized by whether they directly contacted the CALB2 surface ("On CALB2" in magenta) or not ("Off CALB2" in green) (**D–E**). (**J–O**) Representative examples of reconstructed LOC axons. All reconstructions are oriented with the cochlear base to the left and labeled with a colored arrow indicating turn direction (blue: towards the apex; pink: towards the base; yellow: both/bifurcated), the frequency position of its stem and farthest terminal (kHz), and the fraction of Syp puncta On- (magenta) and Off- (green) CALB2+ targets, with the percent of those puncta that are NPY+. (**P**) Peptidergic identity does not correspond to turn direction. Error bars, mean ± SEM. Wilcoxon rank-sum test. (**Q–U**) LOC axon peptidergic identity is not predictive of morphological features or innervation patterns. NPY identity is weakly correlated with axon stem frequency (*r*=0.418, p=0.005) (**Q**), tonotopic span (*r*=0.315, p=0.033) (**R**), and selectivity for CALB2 targets (*r*=0.405, p=0.005) (**U**). There is no correlation between NPY expression and the number of branch points (*r*=–0.284, p=0.056) (**S**) or total number of Syp puncta (*r*=–0.180, p=0.230) (**T**). Color of individual data points corresponds to axonal turn direction. *r*, Pearson's correlation coefficient. n=46 fibers, N=9 animals of both sexes, P27–P30.

The online version of this article includes the following video and figure supplement(s) for figure 5:

**Figure supplement 1.** Additional morphological quantification of OCN terminal axons.

**Figure 5—video 1.** Terminal LOC axon morphology with complex innervation patterns.

https://elifesciences.org/articles/83855/figures#fig5video1

sides of the inner hair cell, although there was a qualitative bias to the modiolar side (*Figure 5B*), similar to what has been reported for LOCs in general (*Hua et al., 2021*; *Liberman et al., 1990*). This distribution confirms that Type I SGN processes are the primary targets of LOCs and argues against selective modulation of specific Type I SGN subtypes by LOC2s.

To quantify the relationship between LOC subtypes and connectivity, we reconstructed individual terminal LOC axons using the same sparse labeling approach that we validated with MOC axon reconstructions (*Figure 4*). Cochlear wholemounts from *Ret^CreER^*; *Igs7^GFP^* animals were immunolabeled for GFP to visualize overall OCN axon morphology, NPY to assess peptidergic identity, Syp to identify putative pre-synaptic sites, and CALB2 to label IHCs and identify Type I SGN subtypes (*Figure 5C–H*, *Figure 5—video 1*). As with the MOC analysis, we were unable to assign projections to specific LOC neurons, and therefore considered each projection independently. Most of the GFP-labeled LOC axons were located in the middle or base of the cochlea, with axons entering the ISB at frequencies ranging from 18 to 69 kHz. This analysis therefore covers much of the ethologically relevant frequency range in mice (*Turner et al., 2005*). NPY distribution was punctate and heterogeneous not just between axons, but also within individual axons, raising the possibility that individual LOC fibers may vary in the signaling molecules released at specific synaptic sites. To assess each LOC axon's potential to release NPY, we used the surface of the axon to mask the Syp and NPY channels (*Figure 5F–H*, *Figure 5—video 1*) and created reconstructions of Syp puncta labeled as either NPY+ or NPY- (open and closed arrowheads, respectively, *Figure 5G and H*). We also identified whether each Syp puncta was in contact with CALB2 immunofluorescence signal (*Figure 5C–E*) to ascertain if either LOC subtype forms selective relationships with Type I SGN subtypes.

Consistent with the known distribution of Type I SGN subtype peripheral processes along the base of the IHC and the corresponding gradient of CALB2 expression (*Figure 5A*), putative synaptic puncta on CALB2+ targets are predominantly positioned on the pillar side of IHCs, and those that do not contact CALB2+ targets are positioned on the modiolar side (*Figure 5E*, *Figure 5—video 1*). Because CALB2 also labels IHCs, it is possible that some Syp puncta contacting CALB2 signal could be LOC contacts onto IHCs. Efferent contacts on mature IHCs have been reported in cat and rat cochleae, although at least in rodents this phenomenon appears to be associated with pathology (*Liberman et al., 1990*; *Liberman, 1980*; *Lauer et al., 2012*; *Zachary and Fuchs, 2015*). However, since most of the LOC Syp puncta in our dataset are positioned below the IHCs, most On-CALB2 Syp puncta are contacting Ia or Ib SGN peripheral processes. Conversely, we can conclude that most Off-CALB2 Syp puncta contact Ic SGN peripheral processes, although a few branches extend away from the SGN peripheral processes and may terminate near supporting cells. While intriguing, interactions with non-SGN targets were too rare for further quantification or analysis.

Our analysis revealed a high degree of variability in LOC axon morphology and connectivity, with no evidence for stereotyped classes. LOC axons turn towards the base, apex, or are bifurcated (indicated by colored arrows in *Figure 5I–O*), and can extend either short, simple branches with few Syp puncta (*Figure 5J and M–O*) or long, complex tangles of branches that form dense forests of Syp puncta (*Figure 5I, K, and L*). Syp puncta were not evenly distributed—some LOC axons can extend upwards of 1 kHz with no Syp puncta (*Figure 5K*). Prior studies of LOC subtypes in rats and guinea pigs suggested bifurcating axons cover longer stretches of the cochlea than axons turning in a single direction (*Warr et al., 1997*; *Brown, 1987*). However, we did not identify any relationship between LOC turn direction and any of the morphological attributes we examined, including the percent of NPY+ Syp puncta, cumulative axon length, tonotopic span, stem frequency, total number of Syp puncta, or the selectivity for CALB2+ SGNs (*Figure 5P*, *Figure 5—figure supplement 1J–N*). Turn direction therefore appears to be a poor predictor for morphological attributes of LOC axons.

Furthermore, we found that no morphological features of the axon—such as its length or branchiness—correlated with its NPY status. Far from falling into distinct categories, the majority of reconstructed LOC axons had both NPY+ and NPY- Syp puncta that were present both on and off CALB2+ targets (e.g. *Figure 5I, K, and L*). Indeed, individual LOC axons can weave from the pillar to modiolar side and extend branches that make pre-synaptic contacts on SGN peripheral processes on both sides of the IHC (*Figure 5E*, *Figure 5—video 1*). In some cases, the Syp puncta along an individual branch preferentially innervate one type of target (e.g. all puncta on CALB2+ targets on the right branch, *Figure 5L*). However, On- and Off-CALB2 Syp puncta were also observed intermingled along the length of an axon (*Figure 5I–K*; *Figure 5L*, left branch). We did identify some axons that had only On-CALB2+ or Off-Calb2+ Syp puncta, but they also showed no relationship with turn direction or the number of NPY+ Syp puncta (*Figure 5M–O*). These terminal axons, although relatively short in length, also varied in the percent of Syp puncta that were NPY+. This was also true of the LOC axon population as a whole, which showed no strong correlation between the percent of NPY+ Syp puncta in an individual LOC and any metric we examined (*Figure 5P–U*, *Figure 5—figure supplement 1O–Q*), including the frequency where the LOC axon enters the ISB (*Figure 5Q*), its tonotopic span (*Figure 5R*), number of branch points (*Figure 5S*), total number of Syp puncta (*Figure 5T*), selectivity for CALB2+ targets (*Figure 5U*), branch depth (*Figure 5—figure supplement 1O*), fiber density (*Figure 5—figure supplement 1P*), or density of Syp puncta (*Figure 5—figure supplement 1Q*). Although some of these measures are weakly correlated with the fraction of NPY+ Syp puncta, NPY identity does not appear to be predictive of LOC morphology or innervation patterns, illustrated most vividly by examining axons with either 0% NPY or 100% NPY+ Syp puncta. On the contrary, even individual LOC axons are poised to have heterogeneous and widespread effects in the cochlea.

## Physiological heterogeneity does not correlate with variability in peptide expression

Given the lack of obvious differences in their innervation patterns, we next asked whether LOC subtypes differ physiologically. Like NPY and Ucn, CGRP (encoded by the *Calca* gene) is expressed as a gradient in LOCs, with high CGRP levels in the medial LSO correlated with expression of NPY and Ucn (*Figure 3O*). We therefore used GFP fluorescence intensity in *Calca-GFP* mice as a proxy for peptidergic identity as a whole. This strategy allowed us to distinguish *Calca*+ LOC neurons from neighboring, *Calca*- LSO principal neurons while also evaluating variability across LOC subtypes. Whole-cell patch-clamp recordings were performed from GFP+ LOC neurons in 200 µm-thick brain slices from 20 P16–P19 *Calca-GFP* mice (*Gong et al., 2003*). At these ages, the intensity of native GFP fluorescence varies in LOC neurons (*Figure 6A*), consistent with differences in *Calca* RNA and CGRP protein levels (*Figure 3J and O*). Although peptide-enriched LOC2s are restricted to the medial wing, cells with variable levels of GFP are also present here (*Figure 6A'*), consistent with the presence of both NPY+ and NPY- LOCs in this region (*Figure 5—figure supplement 1C*). Therefore, cells in the central region of the medial LSO were used for all analyses to reduce the confounding effects of differences that might be related to tonotopic position. Voltage-clamp and current-clamp recordings were assessed for a panel of measures correlating with cell activity and compared across fluorescence intensities of GFP+ cells.

Although the population as a whole was heterogeneous, we observed no systematic differences in the physiological properties of LOCs expressing different levels of GFP (*Figure 6*, *Supplementary*

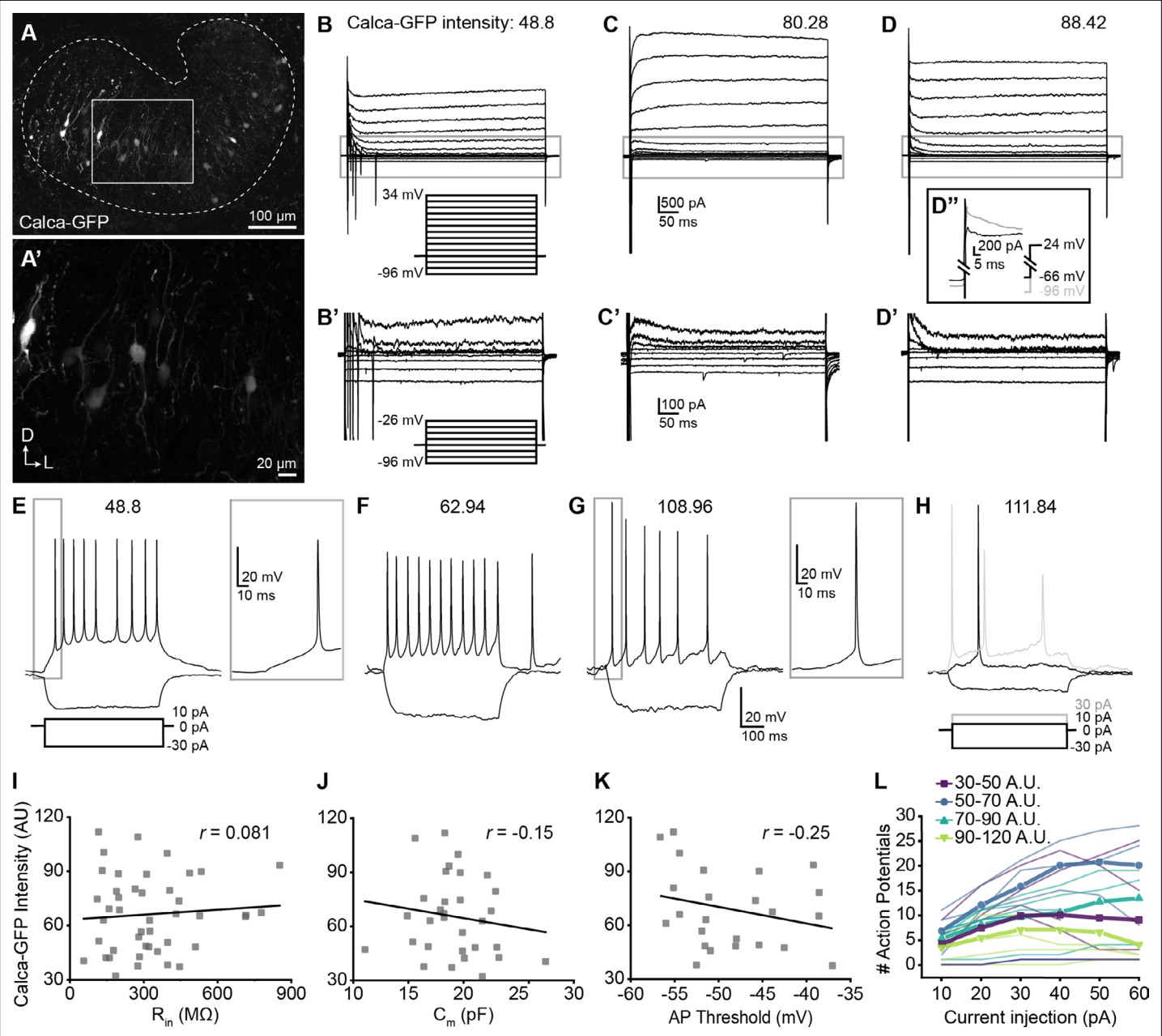

**Figure 6.** Single-cell electrophysiological measures do not correlate with GFP intensity in Calca-GFP+ LOCs. (**A–A'**) Confocal Z-stack of an unfixed brain slice from a *Calca-GFP* mouse showing native GFP fluorescence in greyscale. Dashed line indicates boundaries of the LSO. (**A'**) Zoom of recording region indicated by the box in (**A**). D, dorsal; L, lateral. (**B–D**) Voltage-clamp traces from different LOC neurons in response to the voltage step protocol shown in (**B**, inset). Numbers indicate Calca-GFP fluorescence intensity (arbitrary units, AU). Grey box indicates regions shown in (**B'–D'**). (**B'–D'**) Zoom of responses to steps from –96 to –26 mV from the same traces as shown in (**B–D**), voltage step protocol shown in (**B'**, inset). (**D''**) Inset shows voltage-clamp traces from the same cell as (**D**) in response to a step from –66 mV to 24 mV (black) and from –96 mV to 24 mV (grey) to reveal representative fast-inactivating outward currents. (**E–H**) Current-clamp traces from different LOC neurons in response to the current-step protocol shown in (**E**). The GFP intensity for each cell is indicated above the traces. An additional step to +30 pA is included in (**H**) to show multiple action potentials in response to the current injection (grey). (**I–K**) There is no correlation between Calca-GFP intensity and input resistance ($r$=0.081, p=0.60) (**I**), membrane capacitance ($r$=–0.15, p=0.42) (**J**), or action potential threshold ($r$=–0.25, p=0.26) (**K**). $r$, Pearson's correlation coefficient. (**L**) Input-output curves for current steps from 10 to 60 pA. Thick lines with symbols denote group means for cells of varying Calca-GFP intensities. Thin, transparent lines represent individual cells. GFP intensity is grouped for improved visibility, color code indicated in key.

*file 3*). The input resistance, which was measured during a step from a holding voltage of –66 mV to –76 mV, did not correlate with GFP intensity (*Figure 6I*, *Supplementary file 3*), nor did membrane capacitance measured in voltage-clamp recordings (*Figure 6J*, *Supplementary file 3*). Input resistances were consistent with published work in rat LOCs and putative LOCs in mice (*Sterenborg et al., 2010*; *Fujino et al., 1997*). Voltage-gated currents were assessed by stepping the membrane from a holding potential of –60 mV to voltages between –96 and +34 mV, in 10 mV increments (*Figure 6B–D*). GFP intensity also did not correlate with the magnitude of the outward potassium current measured 485ms following the voltage step to –46 mV or with the steady-state inward current evoked by a step to –96 mV (*Supplementary file 3*). There was heterogeneity in the patterns of potassium currents at steady-state, with some cells exhibiting a slowly increasing outward current during the largest step to +34 mV (*Figure 6B*), some cells having a slowly inactivating outward current (*Figure 6C*), and others exhibiting a fast-onset, non-inactivating outward current yielding a flat waveform (*Figure 6D*). However, all of these groups had similar average GFP intensities (slow increase outward current: 62.71±19.08, n=17; sustained: 67.47±18.36, n=11; decrease: 72.22±25.57, n=14; one-way ANOVA DF = 41, *F*=0.77, p=0.47; mean ± SD).

In some experiments, a 100ms pre-pulse to –96 mV was applied prior to voltage steps that revealed fast-inactivating, likely A-type potassium currents in all of the neurons tested (representative trace, *Figure 6D″*), consistent with enrichment of the voltage-gated potassium channel gene *Kcnd2* in OCNs (*Figure 1—figure supplement 2F*). Following a voltage step from –96 to 24 mV, there was no correlation between A-type potassium current amplitude from the peak to plateau and GFP intensity (*Supplementary file 3*). The decay of the fast-inactivating current was fit to a double exponential, with the fast component likely representing the A-type potassium current. The fast component of the time constant of decay of the fast-inactivating (likely A-type) potassium current did not correlate with GFP intensity (*Supplementary file 3*). Consistent with previous reports, currents indicative of HCN channels were not observed in any LOC neurons (*Sterenborg et al., 2010*). Thus, although LOCs exhibited differences that are consistent with previous characterizations, none of the properties assessed by voltage-clamp correlated with peptidergic identity.

Current-clamp experiments also revealed no correlations between firing properties and GFP intensity (*Figure 6E–H and K–L*). The majority of neurons exhibited action potentials at rest (13 of 23 neurons), some with oscillating patterns as recently reported in LOC neurons (*Hong et al., 2022*). Action potential rates did not correlate with GFP intensity (*Supplementary file 3*). When the membrane was held at ~–63 mV to suppress spontaneous action potentials, there was no correlation between GFP intensity and rheobase, the action potential threshold of the first evoked spike, the amplitude from baseline of the first spike evoked at rheobase, or the number of action potentials evoked by a 10 pA current injection (*Supplementary file 3*). The number of spikes evoked at different depolarizing injection steps between 10 and 60 pA was plotted for each cell to generate input-output curves that indicate the increase in cell activity with increased depolarization. The majority (16 of 19) of cells exhibited multiple spikes in response to the depolarizing steps. There was no relationship between GFP intensity and the slope of the input-output curves generated from each cell (*Figure 6L*, *Supplementary file 3*). A voltage sag in response to hyperpolarizing current steps which indicates I$_h$ currents was not observed in any cells. In summary, although many measures of neuron activity varied among LOC neurons, no measures associated with neuron function correlate with the intensity of GFP and hence LOC subtype identity.

## LOC properties emerge during postnatal development and are modulated by hearing

Since peptide expression seems to be the defining feature of different LOC subtypes, we asked when these differences arise and how they are influenced by hearing. In examining developmental changes in our sequencing data, we found that *Calca*, *Calcb*, *Ucn*, and *NPY* all increase in LOCs during the first four postnatal weeks (*Figure 7A–D*, *Supplementary file 4*). Consistently, analysis of *Npy-GFP* animals (n=3–5 per timepoint) revealed that *Npy-GFP*+ LOC2s are present and localized to the medial wing by at least P5 (*Figure 7E*), with a qualitative increase in the number of NPY+ LOC2s between P5 and P28 (*Figure 7E*, *7—figure supplement 1C–F*). Although there is some variation, cells with the highest *Npy-GFP* levels were always situated in the medial wing of the LSO. In contrast, expression of *Calca* is spatially and temporally dynamic in LOCs across postnatal development, as shown by FISH

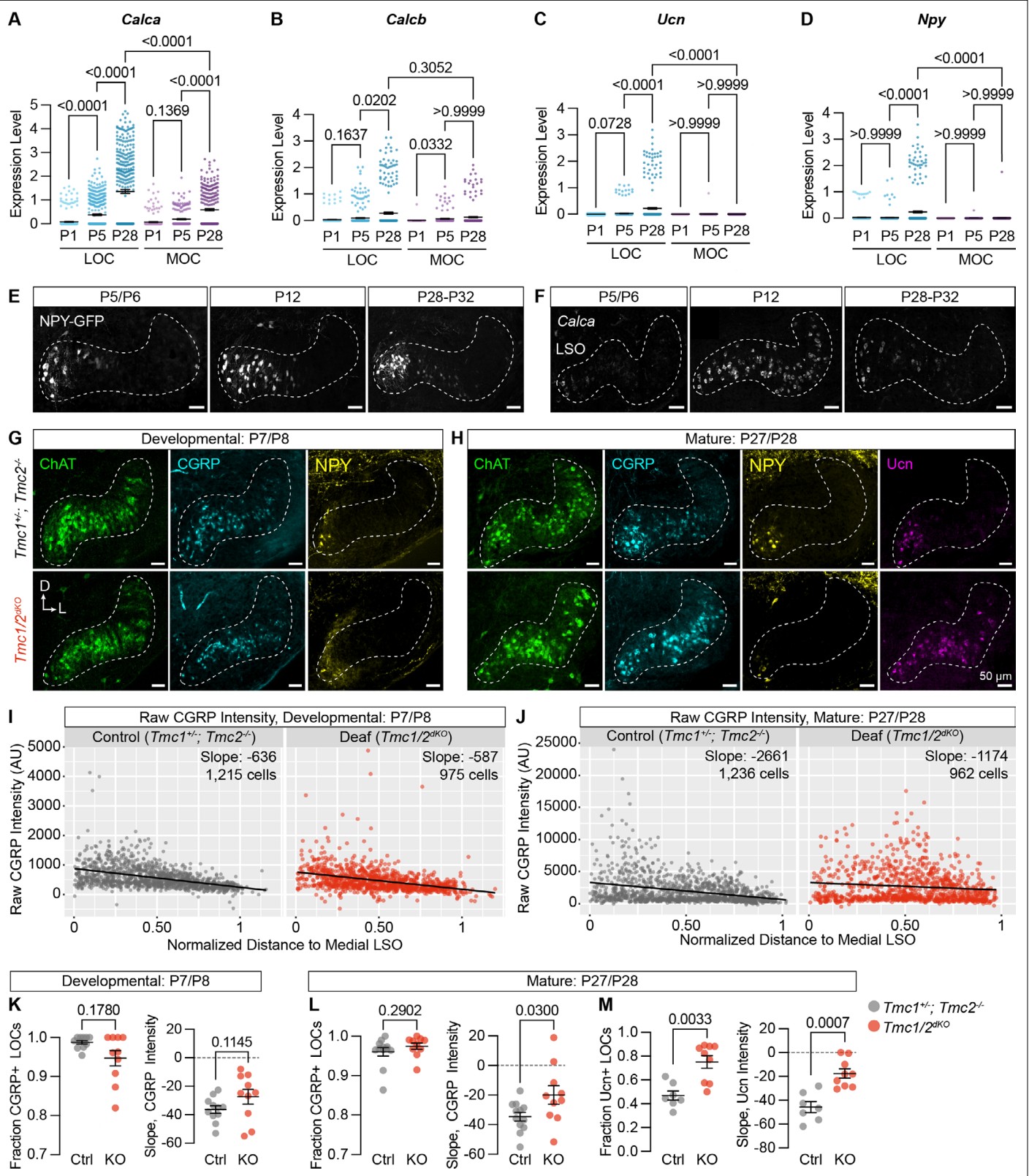

**Figure 7.** Neuropeptide expression in LOCs emerges over postnatal development and is affected by hearing. (**A–D**) Log-normalized counts of *Calca* (**A**), *Calcb* (**B**), *Ucn* (**C**), and *Npy* (**D**) expression across postnatal development as determined by single-nucleus sequencing. Mean ± SEM; Kruskal-Wallis with Dunn's test for multiple comparisons. (**E**) Maximum-intensity projections of confocal z-stacks showing GFP signal in the LSO of *Npy-GFP* mice at varying ages. Representative images are shown from three to five total animals per timepoint. (**F**) Fluorescent *in situ* hybridization of *Calca* expression

*Figure 7 continued*

in the LSO of mice at varying ages. Representative images are shown from two to three animals per timepoint. (**G, H**) Neurotransmitter expression in control (*Tmc1$^{+/-}$; Tmc2$^{-/-}$*) and deaf (*Tmc1$^{-/-}$; Tmc2$^{-/-}$, Tmc1/2$^{dKO}$*) mice before (**G**, P7–P8) and after (**H**, P27–P28) the onset of hearing. Representative images are shown from five to six total animals per age and condition. (**E–H**) Scale bars, 50 μm. Images oriented with dorsal (D) up and lateral (L) to the right. (**I, J**) Quantification of raw CGRP fluorescence intensity for all measured LOCs before (**I**, P7–P8) and after (**J**, P27–P28) the onset of hearing, shown as a function of position along the medial/lateral axis of the LSO. Black line denotes linear regression through all cells. Slope indicates the slope of this regression line. (**K–M**) Expression of CGRP and Ucn in LOC neurons is altered in mature deaf animals. Before the onset of hearing (**K**), the normalized slope of regression lines through the LSO and the relative fraction of CGRP-expressing LOCs does not differ between hearing and deaf animals. After the onset of hearing (**L, M**), the slope of CGRP intensity (**L**) as well as the fraction and slope of Ucn-expressing LOCs is significantly different between hearing and deaf animals. N=7–12 LSOs from four to six animals per condition; Wilcoxon rank-sum test; error bars, mean ± SEM.

The online version of this article includes the following figure supplement(s) for figure 7:

**Figure supplement 1.** Quantification of additional neurotransmitters in *TMC1/2$^{dKO}$* animals.

(2–3 animals per timepoint per probe) (*Figure 7F*). At P5/6, *Calca* levels are low and biased medially. Expression increased dramatically with the onset of hearing, with high levels of transcript detected even in the lateral LSO at P12. By P28, high *Calca* levels are again restricted medially (*Figure 7F*). Previous work identified a similar, transient increase in the expression of Ucn and *Calca* in gerbils and hamsters, respectively, indicating that this phenomenon is conserved across species (*Kaiser et al., 2011*; *Simmons et al., 1997*). Similarly, work in the *Xenopus* lateral line found that peripheral responses to exogenous CGRP increase during development, suggesting that the developmental onset of CGRP expression in efferents may be matched by a gradual increase in expression of CGRP receptors in downstream targets (*Bailey and Sewell, 2000a*). Thus, the LOC2 population seems to be established early in development, but the overall pattern of neuropeptide gene expression is not mature until after the onset of hearing. This finding raises the possibility that peptide status depends on auditory experience.

To assess the impact of hearing on peptide expression in LOCs, we examined expression of Ucn, NPY, tyrosine hydroxylate (TH), and CGRP in the brainstems of constitutively deaf *Tmc1$^{-/-}$; Tmc2$^{-/-}$* double-knockout mice (*Tmc1/2$^{dKO}$*) (*Kawashima et al., 2011*) alongside hearing littermate controls (*Tmc1$^{+/-}$; Tmc2$^{-/-}$*), both before (P7–P8) and after (P27–P28) the onset of hearing. CGRP expression was comparable between deaf and hearing animals at P7–P8 (*Figure 7G, I, and K*; n=11 LSOs from 6 control animals, 10 LSOs from 5 *Tmc1/2$^{dKO}$* animals), assessed both by quantifying the fraction of CGRP+ LOCs and the slope of variation in neurotransmitter levels from the medial to lateral LSO (*Figure 7—figure supplement 1A*). The fraction of NPY-expressing LOCs was also similar between *Tmc1/2$^{dKO}$* animals and controls at P7–P8 (*Figure 7—figure supplement 1C, D*; n=11 LSOs from 6 control animals, 10 LSOs from 5 *Tmc1/2$^{dKO}$* animals). LOCs with the highest level of NPY expression were found in the medial LSO at this timepoint, as well (*Figure 7G*, *Figure 7—figure supplement 1C*), although the slope of NPY expression was quantitatively shallower in *Tmc1/2$^{dKO}$* animals than controls (*Figure 7—figure supplement 1D*).

After the onset of hearing, however, the gradient of CGRP expression was significantly perturbed in deaf mice, with high CGRP levels reaching much farther into the lateral LSO (*Figure 7H, J, and L*; n=12 LSOs from 6 control animals, 10 LSOs from 5 *Tmc1/2$^{dKO}$* animals). Ucn expression was also shifted, reaching farther into the lateral LSO in deaf animals compared to hearing controls (*Figure 7H and M*, *Figure 7—figure supplement 1I*; n=7 control LSOs from 4 animals, 9 *Tmc1/2$^{dKO}$* LSOs from 5 animals). The Ucn antibody was not reliable at P7, so we were unable to evaluate earlier timepoints. In contrast, although NPY levels were qualitatively lower in deaf animals, there were no significant differences in the fraction or slope of NPY expression in adult animals (*Figure 7—figure supplement 1E, F*; n=12 LSOs from 6 control animals, 10 LSOs from 5 *Tmc1/2$^{dKO}$* animals). Differential effects on NPY, Ucn, and CGRP fit with our original observation that peptide expression is heterogeneous in mature LOCs (*Figure 3N and O*, *Supplementary file 1*). TH expression is also reduced in deaf animals compared to controls, consistent with the observation that TH levels can be elevated with sound exposure (*Wu et al., 2020*; *Figure 7—figure supplement 1B, G, H*; n=7 LSOs from 4 control animals, 9 LSOs from 5 *Tmc1/2$^{dKO}$* animals).

These results suggest that peptide expression in LOCs is malleable during postnatal development. To examine whether this flexibility is preserved into adulthood, we exposed 7-week-old C57BL/6J mice to 110 dB 8–16 kHz octave-band noise for 2 hours, a stimulus previously shown to induce TH

expression in mouse LOCs (*Wu et al., 2020*) and induce permanent threshold shifts (*Liberman and Kujawa, 2017*). As expected, the fraction of ChAT+ LOCs expressing TH was elevated 9 days after sound exposure (*Figure 8A–C*; n=4 exposed and 4 unexposed animals of either sex). In addition, we observed an even larger increase in the fraction of LOCs expressing Ucn or NPY (*Figure 8A and D–G*). Because all LOCs already express CGRP, there was no increase in the fraction of CGRP+ LOCs, although the levels of CGRP increased qualitatively throughout the LSO (*Figure 8A, H, I*). Across all neuropeptides, the slope of peptide expression remained unaltered after sound exposure, with higher levels largely constrained to the medial wing (*Figure 8*). In contrast, TH levels increased throughout the LSO, suggesting that the mechanisms driving LOCs to express TH may be different from those that govern neuropeptide expression (*Figure 8A–C*). Thus, hearing shapes the signaling repertoire of a subset of LOC neurons, both developmentally and in adults.

## Discussion

Some variant of the olivocochlear efferent system exists for all hair cell sensory systems: groups of feedback cells project to auditory, vestibular, and lateral line neurons in all classes of vertebrates, and even some invertebrates (*Roberts and Meredith, 1992*). The wide conservation of this circuitry across evolutionary time suggests that OCNs play a crucial functional role. Here, we examined the transcriptional, physiological, and anatomical properties of mammalian OCNs to clarify longstanding questions about their identities and attributes. In doing so, we identified a new, peptide-enriched LOC subtype and found that peptide expression is modulated by hearing, both developmentally and upon acoustic overexposure in adults. Induced peptide expression persists for over a week after noise exposure (*Figure 8*), presenting a mechanism by which LOCs can modulate downstream targets on the timescale of days. Collectively, our findings suggest that a diverse population of OCNs influences the initial detection of sound in a wide-ranging manner that occurs across timescales and changes in response to auditory experience.

### LOC neurons are molecularly, morphologically, and physiologically diverse

Although the entire olivocochlear efferent system is comprised of only a few hundred neurons in mice, we find that these neurons are remarkably heterogeneous, with molecularly distinct cell types that can adopt different states depending on experience. Using single-nucleus sequencing, we identified two transcriptionally distinct subtypes of adult LOCs, which we term LOC1 and LOC2 (*Figure 3*, *Figure 3—figure supplement 1*). The most prominent difference between LOC1s and LOC2s is the expression of genes encoding neuropeptides, including *Calca*, *Calcb*, *Npy*, and *Ucn* (*Figure 3*). Given that LOCs can alter their neurotransmitter expression based on sound exposure (*Niu and Canlon, 2002*; *Wu et al., 2020*; *Figure 8*), it is possible that the transcriptional differences we identified here represent only variations in cell state. However, several lines of evidence support the idea that these clusters represent genuine LOC subtypes. First, LOC1s and LOC2s vary in expression of genes that encode transcription factors and cell-adhesion molecules, attributes that are typically associated with cell types rather than cell states (*Paul et al., 2017*). Second, peptide-high LOC2s are anatomically segregated from LOC1s (*Figure 3O, P*). Moreover, the medial bias in neuropeptide expression is already present by at least P5/P6, prior to the onset of hearing, suggesting that this effect cannot be attributed to a stimulus that might preferentially activate medial LOCs, like ultrasonic vocalizations (*Figure 7E, F*). Finally, although we also show that peptide expression in LOCs can be altered by sound (*Figure 7*, *Figure 7—figure supplement 1*, and *Figure 8*), NPY-expressing cells are biased to the medial LSO across perturbations, whereas cells in the far lateral LSO never express Ucn or NPY (*Figure 7*, *Figure 7—figure supplement 1* and *Figure 8*). Although LOC2s have never been specifically described in the literature, Ucn and CGRP are also enriched in the medial LSO of mature gerbils and rats, respectively, indicating that LOC1/2 differences may be conserved across species (*Kaiser et al., 2011*; *Vetter et al., 1991*).

Historically, classification of LOC neurons in rodents has relied on the anatomical distinction between intrinsic neurons in the body of the LSO and a smaller cohort of shell neurons immediately outside the LSO (*Vetter and Mugnaini, 1992*). Intrinsic and shell neurons have been thought to differ in both their morphology and neurotransmitter expression (*Warr et al., 1997*; *Brown, 1987*; *Darrow*

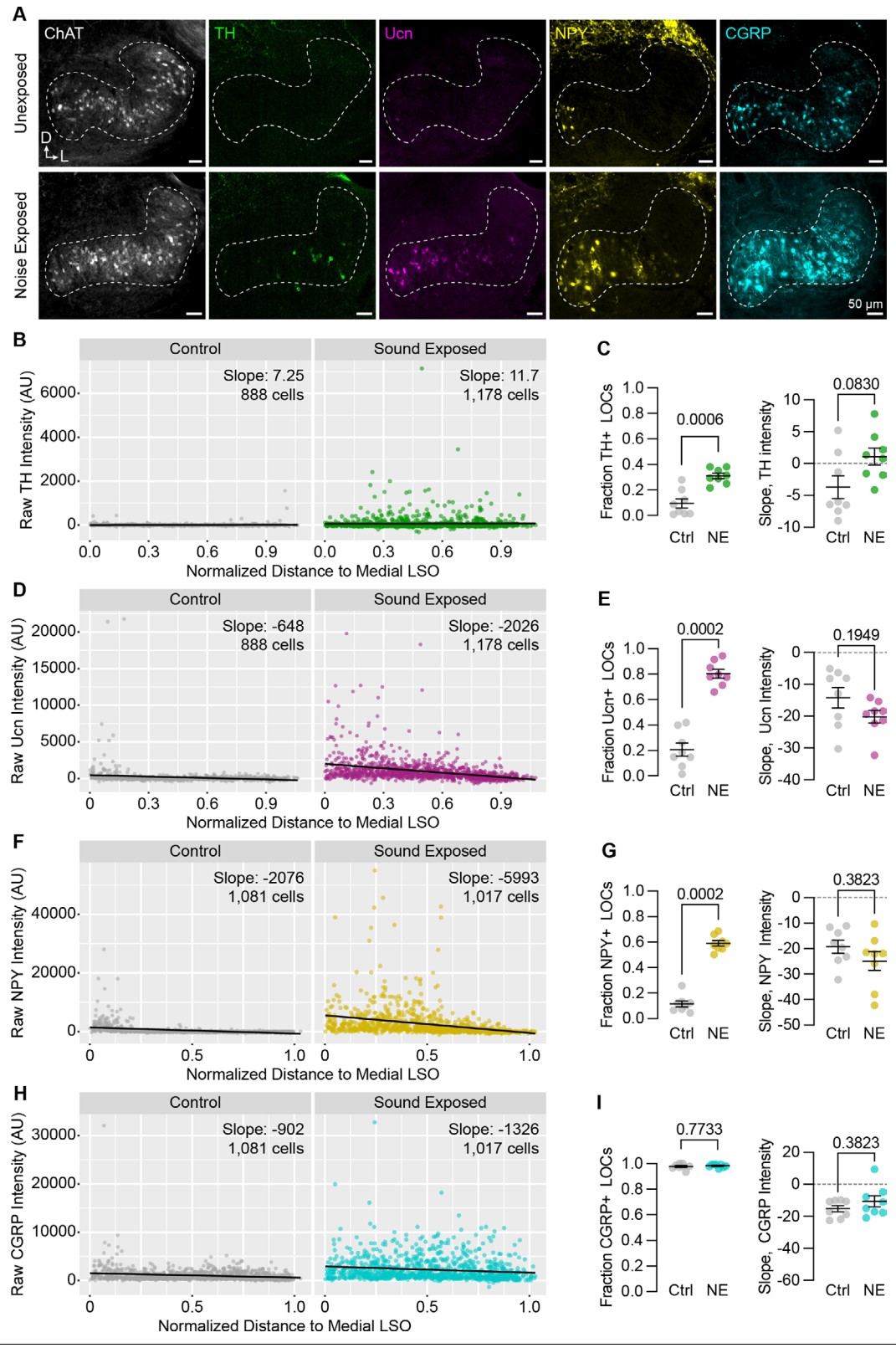

**Figure 8.** Neuropeptide expression in LOCs changes with sound exposure. (**A**) Representative images of ChAT, TH, Ucn, NPY, and CGRP immunolabeling in unexposed (top) and sound-exposed (bottom) animals 9 days after a 2-hr exposure of 110 dB, 8–16 kHz noise. N=4 animals of either sex per condition. D, dorsal; L, lateral. (**B–I**) Quantification of TH (**B, C**); Ucn (**D, E**); NPY (**F, G**); and CGRP (**H, I**) expression. N=8 LSOs from 4 animals per

*Figure 8 continued on next page*

*Figure 8 continued*

condition; 888–1,178 cells. (**B, D, F, H**) Raw fluorescence intensity values of TH (**B**), Ucn (**D**), NPY (**F**), and CGRP (**H**) in annotated LOCs presented as a function of normalized distance to the medial LSO. Black line indicates linear regression of intensity as a function of position within the LSO; Slope denotes the slope of this regression line. (**C, E, G, I**) Quantification of the fraction of LOCs expressing TH (**C**), Ucn (**E**), NPY (**G**), and CGRP (**I**), along with the normalized slope of each protein. Data is presented as mean ± SEM; Wilcoxon rank-sum test.

*et al., 2006b*). At the molecular level, we did not find a population that clearly corresponds to LOC shell neurons, but we cannot rule out that they were too rare to be identified or were excluded from sequencing data because they lack a history of ChAT expression. Nevertheless, although we cannot be certain whether any individual LOC axons in our 3D reconstructions arise from shell or intrinsic neurons, we found that bifurcating axons—a putative marker of shell neuron identity (*Warr et al., 1997*)—did not differ from non-bifurcating axons in any anatomical metric we examined (*Figure 5—figure supplement 1J–N*), indicating that the distinction between shell and intrinsic LOCs does not explain the diversity of LOC morphology present in mice.

The distinction between LOC1s and LOC2s is also insufficient to explain LOC heterogeneity: we found no meaningful correlations between the fraction of NPY+ pre-synaptic puncta in LOC axons and any morphological attributes (*Figure 5*, *Figure 5—figure supplement 1*) or between Calca-GFP intensity and any physiological characteristics (*Figure 6*). Although we are confident that our analysis included both NPY+ and NPY- LOCs (*Figure 5—figure supplement 1C*), we were unable to match reconstructed axons to their cell bodies because our sparse labeling approach captured multiple LOCs per animal. As such, it is possible that the fraction of NPY+ puncta per axon does not reliably correspond to LOC identity based on somatic peptide expression in the LSO. However, even comparison of axons with 0 vs 100% NPY+ puncta revealed no correlations with any morphological property we examined (*Figure 5Q–U*, *Figure 5—figure supplement 1O–Q*). Likewise, electrophysiology experiments revealed diversity in physiological properties that did not track with Calca-GFP intensity (*Figure 6*). It is striking that the molecular differences that define LOC2s—as confirmed by *in situ* hybridization, immunostaining, and genetic labeling—do not correlate with any of the physiological or peripheral morphological characteristics that we measured. One possibility is that our sequencing approach was insufficient to identify all sources of transcriptomic variability that contribute to this diversity. However, these data are consistent with a growing number of studies that find strong correlation between functional, anatomical, and transcriptional properties of major cell types—such as the differences between MOCs and LOCs—and weaker correlation between transcriptional subtypes—such as the differences between LOC1s and LOC2s (*Kim et al., 2019*; *Peng et al., 2021*; *Zeng, 2022*).

## Developmental origins of olivocochlear neurons

Diversification of OCNs begins early, in keeping with the fact that MOCs and LOCs migrate to distinct anatomical locations embryonically (*Frank and Goodrich, 2018*; *Bruce et al., 1997*). Accordingly, MOCs and LOCs can be molecularly distinguished by at least P1, with many cell-type specific markers that persist into adulthood (*Figure 2*, *Figure 2—figure supplement 1*). Historically, MOCs and LOCs were identified only by the anatomical localization of their cell bodies and projections in the cochlea, leading to ambiguity in developmental or pathological conditions where OCNs deviate from their mature configuration (*Zachary and Fuchs, 2015*). The stable, cell-type-specific markers we identified will therefore enable new lines of research into OCN development and function. In addition to cell-type markers like *Col4a4* and *Zfp804a*, we also identified dozens of genes that change during postnatal development (*Figure 2S*, *Figure 2—figure supplement 1H, I*), some of which mirror developmental shifts previously reported in other auditory cell types. For example, both MOCs and LOCs downregulate several GABA receptor subunits while increasing expression of the glycine receptor subunit *Glra1*, indicating a transition from receiving GABAergic to glycinergic inputs (*Korada and Schwartz, 1999*; *Kotak et al., 1998*). More work is needed to determine how these and other suggestive gene expression differences ultimately impact neuronal development and function, starting with confirmation at the protein level.

As with diversification of broader OCN subtypes, LOC1/LOC2 specification may also happen relatively early in development, as LOC2s can be recognized as early as P5 by enriched expression of NPY and CGRP in the medial LSO (*Figure 7E and F*). We find that peptide expression in LOCs

changes during postnatal development in a manner that depends on auditory activity (*Figure 7*), raising the possibility of experience-dependent developmental events (*Figure 7*, *Figure 7—figure supplement 1*). However, developing LOC2s differ from other sensory neurons whose specification relies on activity: some neurons in visual cortex cannot be classified at birth because they express mixed subtype markers, solidifying into mature subtypes only after eye opening (*Cheng et al., 2022*). In contrast, LOC2 markers such as NPY are localized medially both before and after the onset of hearing. Instead, certain attributes of LOCs—namely peptide expression—are malleable based on sensory experience.

## Functional consequences of dynamic neuropeptide expression

LOCs have been reported to express a variety of transmitters, peptides, and modulators, but definitive evidence for many has been lacking. Our data argue against a major role for LOC transmission of opioid peptides (*Safieddine and Eybalin, 1992*; *Vega et al., 2022*), which were detected by neither RNA-seq nor FISH (*Figure 3—figure supplement 1G, H*). On the other hand, our data identify a new transmitter in the LOC repertoire, NPY. The function of these various signaling molecules remains an open question.

Throughout the cochlea, LOC arborizations span large frequency domains and contact multiple SGN subtypes, hinting that their role may be broadly modulatory, perhaps serving to tune sensitivity of SGNs based on behavioral demands (*Guinan, 2018*; *Romero and Trussell, 2022*). Indeed, there is evidence that cues about arousal play this type of role in the vestibular efferent system of fish, possibly to increase the sensitivity of vestibular afferent neurons prior to a rapid movement (*Highstein and Baker, 1985*; *Mathews et al., 2017*). In keeping with this interpretation, LOCs, but not MOCs, express *Hcrtr2,* which encodes an orexin receptor (*Figure 2Q*), suggesting that LOCs may be poised to transmit information about general arousal state to the cochlea. Release of slow-acting, neuromodulatory peptides may be another way that LOCs can alter the general sensitivity of SGNs.

As well as adjusting cochlear function in response to descending cues such as arousal or attention, LOCs may work together with MOCs to protect cochlear circuits from acoustic trauma and other environmental insults (*Fuente, 2015*). Previous work found that the duration and intensity of sound exposure affect the number of LOCs that induce TH expression, with more intense stimuli inducing greater TH expression (*Wu et al., 2020*). We found that not only TH, but also NPY, Ucn, and CGRP are induced after exposure to an intense stimulus (110 dB for 2 hours) that is known to cause noise-induced damage in the cochlea (*Figure 8*; *Liberman and Kujawa, 2017*). It is possible that altered neurotransmitter emission ameliorates the damaging effects of sound exposure. Consistent with this model, recordings from SGNs show that dopamine can reduce SGN firing rates, potentially as a way of reducing excitotoxicity evoked by loud sounds (*Wu et al., 2020*; *Ruel et al., 2001*). Neuropeptides may play a similar role to dampen SGN activity. For example, transcripts of the inhibitory, G-coupled NPY receptor *Npy1r* were detected in low levels in Type Ic SGNs (*Shrestha et al., 2018*). Complicating this hypothesis, most studies on the effects of CGRP in the inner ear of mammals or lateral line of amphibians have found that CGRP plays an excitatory role (*Bailey and Sewell, 2000a*; *Sewell and Starr, 1991*; *Bailey and Sewell, 2000b*; *Maison et al., 2003b*; *Le Prell et al., 2021*). Infusions of CGRP into the guinea pig cochlea increase the amplitude of compound action potentials (CAPs) (*Le Prell et al., 2021*)—a measure of bulk SGN activity—whereas CGRP knock-out mice exhibit decreased auditory brainstem response (ABR) amplitudes (*Maison et al., 2003b*), reinforcing the idea that CGRP enhances SGN firing.

However, SGNs may not be the only target of the various neuropeptides released by LOCs. It has previously been proposed that components of the CRF stress-response pathway, like Ucn, interact with supporting cells in the organ of Corti to mitigate environmental damage, analogous to the way that this signaling pathway promotes the viability of skin (*Graham and Vetter, 2011*; *Slominski et al., 2013*; *Basappa et al., 2010*; *Graham et al., 2011*). In support of this idea, the Ucn receptor *Crhr1* is expressed in cochlear supporting cells, suggesting that they may be an alternate target of LOC signaling (*Graham and Vetter, 2011*; *Liu et al., 2018*). Indeed, we occasionally observe Syp+ puncta on LOC axon branches that extend away from SGN processes into the area where support cells are located. Immune cells are another possible target of LOC peptide release, as NPY and CGRP are both immune-modulatory molecules that could alter inflammatory responses after acoustic injury (*Assas et al., 2014*; *Chen et al., 2020*). The upregulation of Ucn, NPY, and CGRP may therefore serve as a

central cue to stress-response and immune pathways in the ear following injury, similar to the finding that upregulation of TAF4 from mechanosensory neurons promotes recovery from UV damage in the skin (*Hoeffel et al., 2021*). Likewise, the olivocochlear system may serve to both protect cochlear circuits and heighten responses to salient stimuli. Collectively, the molecular, morphological, and physiological diversity described here will inform future efforts to assign functions to LOCs generally and to LOC1s and to LOC2s specifically.

## Materials and methods
### Mice
We used the following mouse strains: *Rosa26^tdTomato* (Ai14; Jax strain 007914) (*Madisen et al., 2010*); *Igs7^GFP* (Ai140D; Jax strain 030220) (*Daigle et al., 2018*); *Rosa26^Sun1-GFP* (Jax strain 021039) (*Mo et al., 2015*); *Chat^Cre* (Jax strain 006410 and 028861) and *Chat^CreΔNeo* (Jax strain 031661) (*Rossi et al., 2011*) *Calca-GFP* (*Gong et al., 2003*); *Npy^FlpO* (Jax strain 030211) (*Daigle et al., 2018*); *Ret^CreER* (*Luo et al., 2009*); *Npy-GFP* (Jax strain 006417) (*van den Pol et al., 2009*); *Tmc1^KO* and *Tmc2^KO* (*Kawashima et al., 2011*); and *Rosa26^FLTG* (Jax strain 026932) (*Plummer et al., 2015*). *Rosa26^tdTomato*, *Igs7^GFP*, *Chat^CreΔNeo*, *Npy^FlpO*, *Npy-GFP*, *Rosa26^FLTG*, and *Tmc1/2^KO* animals were maintained on a C57BL/6 background. *Chat^Cre* and *Rosa26^Sun1-GFP* used for single-cell sequencing experiments were maintained on a mixed background of C57BL/6;129S6. *Calca-GFP* and *Ret^CreER* were on an unknown mixed background of at least 50% CD1. All animal work was conducted in compliance with protocols approved by the Institutional Animal Care and Use Committee at Harvard Medical School or the National Institute on Deafness and Other Communication Disorders Animal Care and Use Committee.

### Sparse labeling of OCNs in Ret^CreER mice
Timed-pregnant *Ret^CreER* female mice, crossed to *Igs7^GFP* male mice, were given 0.8–1.3 mg tamoxifen (Sigma Cat#T5648-1G)/0.8–1.3 μg β-estradiol (Sigma Cat#E2758-1G) in corn oil (Sigma Cat# C8267-500ml) via oral gavage at embryonic day 16.5–17.5. Tissue was harvested from the *Ret^CreER*;*Igs7^GFP* offspring at P27–P30.

### Noise exposure
Noise exposure occurred in a custom plexiglass trapezoidal box located inside a tabletop noise-proof chamber. Acoustic stimuli were delivered from a speaker at the top of the chamber. To ensure even sound exposure, animals were placed in individual wire mesh boxes on a mesh platform located in the center of the chamber. Sound pressure levels were measured with a ¼" free-field microphone (PCB 378C01) which was calibrated prior to each exposure session (Larson-Davis CAL200). Stimuli consisted of an 8–16 kHz octave-band noise with a mean intensity of 110.2±0.716 dB SPL (absolute range). Four animals were exposed at a time.

### Single-nucleus sequencing
For P26–P28 datasets, animals were *Chat^Cre*; *Rosa26^Sun1-GFP*. For P1 and P5 collections, animals were *Chat^CreΔNeo*; *Rosa26^Sun1-GFP*. Each individual collection contained pooled tissue from 5–11 animals of both sexes. In total, collections included the following: P1, 8 males and 5 females; P5, 7 males and 9 females; P26–P28, 20 males and 12 females. For developmental datasets, we collected all animals in each of two independent litters at each timepoint. For adult datasets, we collected all animals from each of four independent litters to accommodate lower yields from the heavily myelinated adult brainstem. All animals were on a mixed background of C57BL/6J;129S6. First, we dissected tissue from the ventral brainstem. In adults, we collected roughly the ventral half of the brainstem, in a fragment of the hindbrain that included the entire SOC as well as the FMNs (*Figure 1C*). In P1 and P5 animals, we used a brain matrix (Zivic Instruments BSMNS001-1) to align the brains, then removed a segment of approximately 2 mm along the rostral-caudal axis, beginning at the front of the brainstem and extending caudally to include the FMNs. When it was possible to do so without damaging the tissue, we also removed the dorsal portion of the brain.

Tissue was dissected into an ice-cold buffer solution containing 0.25 M sucrose, 25 mM KCl, 5 mM MgCl$_2$, 1 M Tricine-KOH, 1 μM tetrodotoxin (TTX, Cayman Chemical 14964), 50 μM APV (Tocris 0106), and 20 μM DNQX (Sigma D0540). Dissected brain tissue was pooled and placed into a dounce

homogenizer containing 25 mM KCl, 5 mM MgCl$_2$, 1 M Tricine-KOH, 1 mM DTT (Sigma D0632), 150 µM spermine tetrahydrochloride (Sigma S1141), 500 µM spermidine trihydrochloride (Sigma S2501), 80 u/mL RNAsin Plus RNase Inhibitor (Promega N2615), and one tablet of protease inhibitor cocktail (Roche 11836170001). Midway through homogenization, we added IGEPAL CA-630 (Sigma I8896) to a final concentration of 0.32%. Tissue was homogenized until no visible chunks of tissue remained, then filtered through a 40 µm cell strainer. Next, we added 5 mL of 50% iodixanol (OptiPrep Density Medium, Sigma D1556) with 7.5 mM KCl, 1.5 mM MgCl2, 6 mM Tricine-KOH, pH 7.8, and 80 u/mL RNAsin. Next,~9 mL of homogenate was added to a density gradient of 30% and 40% iodixanol and spun for 25 minutes at 10,000 g at 4 °C. After removing most of the top layer, we extracted 400 µL from the interface of the 30% and 40% iodixanol layers. Next, we added 600 µL 1% BSA in PBS and 10 µL of Draq7 (Abcam) to stain the dissociated nuclei. Finally, cells were filtered through a 40 µm Flowmi cell strainer (Sigma-Aldrich).

To isolate Sun1-GFP-positive cholinergic nuclei, nuclei were sorted on a BD FACSAria II (BD Biosciences) by the Harvard Immunology Flow Cytometry Core. After FAC-sorting, nuclei were spun at 4 °C for 5 min at 750 rpm and re-suspended in ~40 µL of 1% BSA in PBS. After estimating concentration by counting DAPI-labeled nuclei on a hemocytometer, nuclei were loaded into a single-cell 3′ chip from 10x Genomics, following the manufacturer's directions. If the estimated concentration was greater than 300 nuclei/µL, we divided the sample in half and spread it across two lanes of the 10x Genomics microfluidics chip in order to reduce the proportion of multiplets. Otherwise, we loaded the maximum volume of the sample onto the chip. Mature datasets from P26–P28 animals were processed with the Chromium single-cell 3′ library and gel bead kit v2 (10x Genomics, PN-120267); developmental datasets from P1 and P5 animals were processed with the Chromium single cell 3′ GEM, library, & gel bead kit v3 (10x Genomics, PN-1000092). cDNA libraries were generated according to the manufacturer's directions. The final libraries were sequenced on an Illumina NextSeq 500 by the Harvard Bauer Core (75 cycle kit).

## RNAseq analysis

### Alignment

Raw reads were converted to fastq files using the cellranger pipeline from 10x Genomics, v. 2.1.0–3.0.1. All datasets were aligned to a modified version of the mm10–3.0.0 reference transcriptome using cellranger v. 3.0.2. Because nuclei contain large amounts of unprocessed RNA, the standard mm10 reference transcriptome was altered so that reads were aligned to all annotated transcripts, rather than aligning exclusively to exons.

### Filtering and exclusion criteria

Following alignment, data was imported into R (v. 3.5.3–4.0.5) and analyzed with the Seurat package (v. 3.1.4–3.2.3; *Stuart et al., 2019*). Initially, each individual library from independent 10x lanes was processed separately to remove low-quality cells and genes that were detected infrequently. Each library was filtered to remove genes that were not detected in at least 3 cells and cells that contained fewer than 500 unique genes. Each library was further filtered to discard any cell in which more than 1% of detected genes mapped to mitochondrial DNA. Because we sequenced individual nuclei rather than single cells, the presence of notable amounts of mitochondrial genes likely indicates contamination of RNA from outside the nucleus or other issues with sample quality (*Ilicic et al., 2016*; *Luecken and Theis, 2019*). The average mitochondrial content of our libraries was less than 0.45% in each dataset.

Because libraries from the P1 and P5 collections were prepared with the v3 kit, they had far higher nUMI and nGenes than the mature datasets, which were collected with the 10x v2 kit (median nUMI for developmental datasets was 6,157–10,171; for adult datasets, 1,935–3,978; *Figure 1—figure supplement 1*). We therefore established different filtering criteria for the developmental and adult data: we kept developmental cells that had more than 1,000 detected UMI and more than 750 genes and adult cells that expressed more than 750 UMI and more than 500 genes. In addition, we excluded some potential multiplets by removing all cells for which the nUMI or nGenes detected was more than two standard deviations away from the mean. We chose a standard deviation-based cutoff rather than a specific upper limit for nUMI or nGenes because the distribution of detected genes and UMI differed between datasets based on variability in sequencing depth.

## Normalization and batch correction

After filtering low-quality cells, all libraries generated from parallel collections were merged prior to normalization (that is, all libraries that were collected simultaneously but distributed over multiple lanes of the 10x microfluidics chip). Data was normalized by fitting the gene counts to a regularized binomial regression function, implemented in the scTransform package for Seurat (*Hafemeister and Satija, 2019*). Next, we accounted for batch effects and variability between developmental and adult data using a CCA-based integration method implemented in Seurat v. 3 (*Stuart et al., 2019*). We used 3000 variable genes and 30 dimensions in CCA space for the integration, consistent with previous applications of this method.

After integrating the data, we normalized gene counts by dividing the counts for each gene by the total counts for each cell. These values were then multiplied by 10,000 and natural-log transformed. For visualizing counts on heatmaps, data for each gene was centered around its mean value and scaled by dividing by the gene's standard deviation.

## Clustering

To cluster the data, we performed principal component analysis (PCA) on the integration vectors produced in the previous steps. We used the top 21 PCs as input to the clustering algorithm based on the relative amount of variance explained by each PC, as visualized on an elbow plot. Finally, we clustered the data using a graph-based clustering algorithm implemented with the FindNeighbors and FindClusters algorithms in Seurat (*Stuart et al., 2019*; *Waltman and van Eck, 2013*), using a resolution of 0.8. The number and proportion of MOCs and LOCs was stable across a range of input parameters, varying by only ±2 cells from a resolution of 0.4–1.6 and input dimensions of 16–30 PCs.

## Differential expression analysis

Differential expression analysis was performed using a non-parametric Wilcoxon rank-sum test on log-normalized counts data, implemented with the FindMarkers function in Seurat. Post-hoc adjustments were performed with a Bonferroni correction based on all genes in the dataset.

## Sub-clustering of adult OCNs

To analyze subsets of adult OCNs, data was filtered and normalized as described above. However, in order to focus specifically on subtypes found in mature OCNs, batch correction and integration was applied only to the datasets from P26–P28 animals. After clustering the adult cell types as described above, we identified OCN clusters based on their co-expression of *Gata3* and motor neuron markers like *Isl1* and *Tbx20*. We then subset the data to include only cells in those two OCN clusters and repeated the clustering analysis. The counts data was then log-normalized and scaled prior to differential expression analysis and visualization.

## Geneset analysis

Geneset enrichment analysis to identify specific categories of genes expressed in various cell populations was based on previously curated gene lists (*Shrestha et al., 2018*; *Paul et al., 2017*) and annotations from the HUGO Gene Nomenclature Committee (HGNC, genenames.org). In total, these genesets included 280 guidance and adhesion molecules, 215 ion channels, 215 neurotransmitter receptors, 1,634 transcription factors, and 120 genes involved in neuronal signaling or neurotransmitter or neuropeptide synthesis.

## Fluorescent in situ hybridization

P27–P28 mice on a C57BL/6J background were perfused with ice-cold, RNAse-free 4% paraformaldehyde (PFA, Electron Microscopy Sciences) in 1x Sorenson's buffer. Brains were post-fixed overnight in 4% PFA at 4 °C and cryopreserved in sucrose prior to cryoembedding in NEG-50 (Epredia) and sectioning at 20 μm on a cryostat (Leica). Prior to staining, slides were brought to room temperature and incubated at 50–55°C for 15 minutes. Next, sections were fixed in 4% PFA in 1x Sorenson's buffer for 10 minutes on ice, then rinsed twice in phosphate-buffered saline (PBS) with 0.1% Triton X-100 (Sigma T9284; PBST). Slides were then treated with 1 μg/mL Proteinase K (Sigma P-6556) for 10 minutes at room temperature, rinsed twice with PBST, and fixed again in 4% PFA on ice. After

rinsing twice with PBST, slides were run through an ethanol dehydration gradient of 50%, 70%, and 100% ethanol. After drying, the HCR in-situ protocol was followed as described (*Choi et al., 2018*). The following probes were synthesized by Molecular Instruments, Inc and were used in the indicated concentrations: *Cadps2*, 2 pM; *Calca*, 1 pM; *Col4a4*, 2 pM; *Zfp804a*, 2 pM. Additional probes were synthesized by IDT and used in the following concentrations: *Penk*, 2 pM; *Pdyn*, 1 pM. Sequences for custom probes from IDT are in *Supplementary file 2*. All samples were imaged on either a Leica SP8 or Zeiss LSM800 confocal microscope. Because we expected minimal inter-animal variability in the expression of cell-type markers, all FISH experiments used tissue from 2 to 3 animals per probe.

## Immunofluorescence
### Brainstem
Animals were perfused with ice-cold 4% PFA in PBS. Brains were removed, post-fixed overnight at 4 °C in 4% PFA, and rinsed 3 times in PBS. Prior to sectioning, brains were embedded in a BSA-gelatin mixture consisting of 0.4% gelatin, 23.3% BSA, 5.9% formalin, and 0.32% glutaraldehyde. Brains were then sectioned at 40–50 µm on a Leica vibratome. Sections were blocked and permeabilized in 10% normal donkey serum (NDS) with 0.4% Triton X-100 in PBS for 1–3 hr. Antibodies were diluted in a buffer consisting of 5% NDS and 0.4% Triton X-100 in PBS and incubated for 1–2 nights at room temperature. The following primary antibody concentrations were used: Mouse anti-CGRP (Abcam, 1:500–1:1,000); Rabbit anti-NPY (Cell Signaling, 1:500); Goat anti-ChAT (Millipore Sigma, 1:500); Rabbit anti-Ucn (Sigma-Aldrich, 1:500); Chicken anti-TH (Abcam, 1:1,000); Chicken anti-GFP (Aves, 1:2,000). Sections were rinsed 3 times with PBS, then incubated in secondary antibodies for 1.5–3 hr at room temperature. All secondary antibodies were used at 1:1,000 in 5% NDS and 0.4% Triton X-100 in PBS (Key Resources Table). Finally, sections were incubated in DAPI (Invitrogen, 1:10,000) for 10 minutes at room temperature, rinsed twice with PBS, and mounted with Vectashield hard-set medium (Vector Labs). Samples were imaged on either an Olympus VS120 fluorescent microscope or a Zeiss LSM 800 confocal microscope.

### Cochlea
Animals were perfused with 4% PFA in PBS. Temporal bones were removed and a small hole was made in the dorsal side of the bone before post-fixing overnight at 4 °C in 4% PFA. After rinsing 3 times in PBS, temporal bones were decalcified with 120 mM EDTA for 48–72 hr at 4 °C, then rinsed three times in PBS. Cochlea were dissected from the decalcified temporal bones and micro-dissected into three or four turns. Cochlear turns were blocked and permeabilized in 5% NDS with 1% Triton X-100 in PBS for 45–90 min at room temperature. Primary antibodies were diluted in a buffer consisting of 1% NDS and 0.5% Triton X-100 in PBS and incubated for 1 night at 37 °C. The following primary antibody concentrations were used: Mouse anti-Synaptophysin (Synaptic Systems, 1:600), Rabbit anti-NPY (Cell Signaling, 1:1000), Goat anti-CALB2 (Swant, 1:1,000), Chicken anti-GFP (Aves, 1:2,000), Guinea Pig anti-Parvalbumin (Synaptic Systems, 1:1,000). The following day, tissue was rinsed three times with 0.5% Triton X-100 in PBS before incubating with secondary antibodies in 1% NDS and 0.5% Triton X-100 in PBS at 37 °C for 2–4 hr. The following secondary antibodies were used at 1:1,000: Donkey anti-Goat Dylight 405, Donkey anti-Rabbit Alexa 647, Donkey anti-Chicken Alexa 488, Donkey anti-Guinea Pig Alexa 647, Donkey anti-Mouse Alexa 568. Tissue was then rinsed two to three times with 0.5% Triton X-100 in PBS, then two times with PBS before mounting with Fluoromount-G (Southern Biotech) on slides with a 0.5 mm spacer between the slide and coverslip. Cochlear wholemounts were imaged on a Leica SP8 or Zeiss LSM 800 confocal microscope.

## Image analysis
### Cochlea
Low-magnification images were acquired of all turns for each cochlea and imported into Fiji (*Schindelin et al., 2012*) with the Bio Formats plugin (*Linkert et al., 2010*). The measure_line plugin was used to map the cochlear length to cochlear frequency (Key Resource Table). High-magnification z-stack images were acquired using either a 40× objective on a Leica SP8 confocal or a 40× or 63× oil-immersion objective on a Zeiss LSM 800 confocal. In a subset of cases in which the labeled OCN axon was very thin, the z-stack image was acquired using Airyscan 2 imaging and processing on a Zeiss LSM 800 confocal.

## Reconstruction of sparsely labeled OCN axons in the cochlea

Z-stack images of sparsely labeled OCN axons in *Ret^CreER*; *Igs7^GFP* cochlea were imported into Imaris (Bitplane AG, v. 9.8–9.9.0) for reconstructions of sparsely labeled OCN axons. To capture the full range of OCN morphologies, as well as any possible sex-specific differences, 4–5 animals of each sex were collected for this analysis. All traceable OCN fibers from these nine *Ret^CreER*; *Igs7^GFP* animals were reconstructed and used for morphological analysis. The filament function was used to semi-automatically reconstruct individual axons based on the GFP fluorescence signal. MOC axons were traced starting in the OSL and LOC axon reconstructions began at the habenula in order to measure total filament length in the ISB consistently across axons. The surface function was used to create surfaces of OCN axons (from the GFP fluorescence) and CALB2+ SGN fibers and IHCs. The surface of the OCN axons (based on the GFP fluorescence signal) was used to mask the Syp and NPY fluorescence channels. From those masked channels, the surface function was again used to create Syp and NPY surfaces. The Syp and NPY puncta surfaces were created from a surface grain size of 0.115 µm and those larger than ~1.6 µm were divided. All surfaces were manually reviewed and adjusted as needed; for example, for the masked Syp and NPY channels, any puncta that was cut off by the masking surface was reviewed and excluded if it was not fully colocalized with the GFP channel. Syp surfaces were then filtered based on colocalization with NPY (i.e., proximity ≤0) to identify NPY+ and NPY- Syp puncta. These surfaces were subsequently filtered by proximity to the CALB2 surface to identify On- and Off-CALB2 Syp puncta. Filament and surface statistics were exported and analyzed in Excel, R, and GraphPad Prism.

## Manual quantification of somatic OCN peptide expression

For initial quantification of NPY and Ucn co-expression (*Figure 3*, *Supplementary file 1*), confocal images were scored by hand for overlapping expression of neuropeptides. Visible signal was attributed to a Ucn- or NPY-expressing LOC if it had a morphology consistent with LOC neurons and was located within the anatomical bounds of the LSO. Cells were categorized as co-expressing Ucn and NPY if signal from both channels overlapped completely in all three dimensions.

## Automated quantification of somatic peptide expression

To extract quantitative differences in protein expression throughout the LSO, we established a semi-automated segmentation and quantification pipeline. After imaging, 1–3 LOC sections per hemisphere were selected that included LOCs in both the medial and lateral extremes of the LSO. Individual images were excluded if cells could not be confidently segmented due to poor staining quality or optical aberrations. For CGRP and NPY quantification in mature Tmc knock-out animals, the final dataset consisted of the following: 24 images of 12 LSOs from 6 control animals; 18 images of 10 LSOs from 5 double-knockout animals. The Ucn and TH quantification of mature Tmc knockout animals included the following: 15 images of 7 LSOs from 4 control animals; 17 images of 9 LSOs from 5 double-knockout animals. Quantification of CGRP and NPY expression in P7/P8 Tmc knockout animals was derived from 19 images of 11 LSOs from 6 control animals and 15 images of 10 LSOs from 5 double-knockout animals (*Figure 7*, *Figure 7—figure supplement 1*). Quantification of Ucn and TH intensity after sound exposure (*Figure 8*) was derived from 16 images of 8 LSOs from 4 unexposed animals and 15 images from 8 LSOs of 4 sound-exposed animals. Quantification of CGRP and NPY intensity was extracted from 15 images of 8 LSOs from 4 animals for both exposed and unexposed conditions. All quantification experiments included 4–6 animals per genotype or condition. All experiments and conditions included pooled animals from both sexes. In all cases, analysis was conducted with experimenter blind to genotype or noise-exposure status.

Images were pre-processed for segmentation by enhancing local contrast using a Gaussian filter-based local normalization algorithm as previously described (*Zhang et al., 2021*) (see Key Resources Table). Resulting images were automatically segmented using CellPose v. 1.0 (*Stringer et al., 2021*). Resulting soma masks were manually curated downstream of automated segmentation to meet the following criteria: (**a**) objects were located within the anatomical boundaries of the LSO; (**b**) objects were labeled by either CGRP or ChAT; (**c**) objects were cell shaped (*Figure 7*, *Figure 7—figure supplement 1A*). Neuropil masks were then obtained by dilating the soma masks by 20 pixels and subtracting 5-pixel-dilated soma masks. Soma and neuropil masks were then applied to unprocessed images to

extract mean pixel fluorescence intensity. Finally, mean neuropil pixel intensity was subtracted from the mean intensity of the affiliated soma to account for variability in local background fluorescence.

Both the medial-lateral orientation and length of each LSO were annotated manually by drawing a line through the LSO in each image (*Figure 7*, *Figure 7—figure supplement 1A*). The length of each LSO was normalized to 1 based on this line in order to facilitate comparisons of relative LOC position within the LSO across samples. To compute the fraction of cells expressing each protein, a threshold for each protein was selected using raw fluorescence intensities that consistently yielded cell count fractions consistent with manual counts. To calculate the slope of protein expression across each LSO, minimum intensity values of all cells were restricted to 0 (that is, any negative intensity values were set to 0). Next, cells in each LSO were normalized by dividing the fluorescence intensity of each cell by the intensity of the brightest cell in that LSO. A linear regression model based on the relationship between cell position and normalized fluorescence intensity was calculated in R. The slope of this regression line was then extracted from each LSO. Slopes of protein expression and the fraction of cells expressing each protein between groups were imported to GraphPad prism and assessed for statistical significance with a Wilcoxon rank-sum test between groups. Data is shown as mean ± standard error of the mean (SEM) (*Figures 7 and 8*, *Figure 7—figure supplement 1*).

## Electrophysiology

### Slice preparation

Brain slices were prepared from 20 P16–19 *Calca-GFP* mice of either sex. Sample sizes are consistent with standard practice for recording from brainstem auditory neurons (*Torres Cadenas et al., 2020*). Mice were euthanized by carbon dioxide inhalation at a rate of 20% of chamber volume per minute, then decapitated. The brain was removed in cold artificial cerebrospinal fluid (aCSF) containing (in mM): 124 NaCl, 2 $CaCl_2$, 1.3 $MgSO_4$, 5 KCl, 26 $NaHCO_3$, 1.25 $KH_2PO_4$, 10 dextrose. 1 mM kynurenic acid (KA) was included during slice preparation. A subset of experiments was performed in aCSF containing the following: 126 NaCl, 2 $CaCl_2$, 1.3 $MgSO_4$, 2.5 KCl, 26 $NaHCO_3$, 1.25 $KH_2PO_4$, 10 dextrose. The pH was 7.4 when bubbled with 95% $O_2$/5% $CO_2$. 200 μm coronal brain slices containing nuclei of the SOC including the MNTB and VNTB were cut with a vibratome (Leica VT1200S) in cold aCSF +KA. Slices were stored in a custom interface chamber at 32 °C for one hour and then allowed to recover at room temperature until time of recording. Recordings were performed at 35 ± 1 °C. Slices were used within four hours of preparation.

### Patch-clamp electrophysiological recordings

Brain slices were transferred to a recording chamber continuously perfused at a rate of ~2–3 mL/min with aCSF bubbled with 95% $O_2$/5% $CO_2$. The slices were viewed using a Nikon FN-1 microscope with DIC optics and a Nikon NIR Apo 40 X/ 0.80 N/A water-immersion objective. The images were collected as Mono 8-bit.nd2 files with a QICLICK, MONO 12BIT, non-cooled camera (Nikon) or Mono 16-bit. nd2 files with a CoolSNAP DYNO camera (photometrics) and viewed using NIS-Elements (Nikon). LOC neurons were identified for recordings by their position in the medial arm of the LSO and visibility using green epifluorescence (488 nm emission filter, Lumencor Sola lamp). DIC and fluorescent images were obtained from each neuron prior to recordings. The recordings were performed using a MultiClamp 700B and DigiData 1440 A controlled by Clampex 10.6 software (Molecular Devices) or a HEKA EPC 10 USB Double Patch Clamp Amplifier controlled by Patchmaster Next Software (HEKA Electronik), sampled at 50 kHz and filtered on-line at 10 kHz. The internal solution contained (in mM): 125 K-gluconate, 5 KCl, 1 $MgCl_2$, 0.1 $CaCl_2$, 10 HEPES, 1 EGTA, 0.3 Na-GTP, 2 Mg-ATP, 1 $Na_2$-phosphocreatine, 0.25% biocytin and 0.01 AlexaFluor-594 hydrazide. The pH was adjusted to 7.2 with KOH.

Recording pipettes were pulled from 1.5 mm outer diameter borosilicate glass (Sutter Instruments) to resistances of 3–6 MΩ. Series resistances were corrected more than 60%. The cells were voltage-clamped at –66 mV (liquid junction potential corrected) unless stated otherwise. I-clamp values were corrected for a liquid junction potential of –6 mV. Chemicals were obtained from Fisher Scientific or Millipore Sigma.

Prior to breaking into the cell, a low-magnification image of each LSO was taken using 488 nm filters under epifluorescent illumination. High magnification images of each cell were used to obtain maximum-intensity information of GFP signal, upon which a post-hoc correction was applied for both

the number of light flashes that the slice had been exposed to for multiple recordings in the same slice, and for the depth of the cell in the tissue.

## Electrophysiology: Statistical analysis

Voltage- and current-clamp recordings were analyzed in Clampfit 10.6 (Molecular Devices) or MATLAB 2020a (Mathworks). The intensity of each cell was obtained from an image taken immediately prior to performing patch-clamp recordings. The Auto Detect Area tool of NIS-Elements AR (version 4.50) was used to define a region of interest (ROI) around each cell. Detection tolerance was adjusted as necessary and on rare occasions the ROI had to be hand drawn using the Polygon tool. The maximum intensity within the ROI was obtained using the 'MaxIntensity' measurement feature, which is the maximal of pixel intensity values derived from the intensity histogram. Multiple cells were recorded from each brain slice, and cells were filled with red fluorescent tracer to prevent multiple recordings from the same cell. To compensate for bleaching due to repeated exposure to the fluorescent lamp, in separate experiments cells were imaged multiple times to replicate recording and illumination conditions. A correction factor based on the formula of the line fit to the decrement in intensity over successive illuminations was then applied to individual neurons based on the number of times that each slice was illuminated. Similarly, the intensity was compensated for the depth of the cell in the slice, which was measured during each recording using the calibrated micromanipulators used for placing the patch-pipette (Sutter Instruments). Corrected intensity values ranged from 32.24 to 111.8. Action potential threshold was manually measured at the clear beginning of the rising phase of the voltage response. Input-output curves were calculated from the linear portion of the curve prior to a plateau in the number of evoked action potentials.

All statistical analyses were performed using Origin v2021 (Origin Laboratories). A correlation between patch-clamp recording derived measures of cell function and cell intensity were tested with a single linear regression fit to the data using the 'Fit linear' function in Origin. A relationship was considered significant if the linear fit had a p-value <0.05. For grouped data, the intensity differences of cells in each group were tested with a one-Way ANOVA. For data presentation, voltage and current-clamp traces were low-pass filtered at 2 kHz. Data are presented as mean ± standard deviation (*Supplementary file 3*).

## Resource availability

### Lead contact

Further information and requests for resources and reagents should be directed to and will be fulfilled by the lead contact, Lisa Goodrich (lisa_goodrich@hms.harvard.edu).

## Materials availability

Sequences for *Penk* and *Pdyn* in-situ probes are available in *Supplementary file 2*. Custom Molecular Instruments probes for *Cadps2*, *Col4a4*, and *Zfp804a* are available from the manufacturer.

## Acknowledgements

We are grateful to Lauren Kreeger, Charlie Liberman, Gabriel Romero, and all members of the Goodrich and Weisz labs for insightful discussion and support, and to Matt McGinley for his feedback and attention to the manuscript. Stephen X Zhang provided assistance with automated image analysis and helpful comments on the manuscript. Sadie Schlabach provided technical assistance with portions of the histology presented in this paper. We are grateful to David Ginty for providing *Ret*^CreER and *Calca-GFP* mouse lines, Gordon Fishell and Michael Greenberg for providing the *Rosa26*^Sun1-GFP mouse line, and David Corey for providing *Tmc1*^KO and *Tmc2*^KO mouse lines. We thank Bernardo Sabatini for the use of the 10x controller. Sandy Nandagopal and Natasha O'Brown provided sequences for *Penk* and *Pdyn* HCR probes. We are grateful to Ishmael Stefanov-Wagner, Ken Hancock, and Evan Foss of the Engineering Core at the Eaton-Peabody Laboratories of Mass Eye and Ear for assistance setting up our sound-exposure system. We thank the Neurobiology Department and the Neurobiology Imaging Facility for consultation and instrument availability that supported this work. This facility is supported in part by the HMS/BCH Center for Neuroscience Research as part of an NINDS P30 Core Center grant NS072030. Sequencing for this project was performed by The Bauer Core Facility

at Harvard University, along with library quantification and quality-control tests. Additional library quantification was performed by the Biopolymers Facility at Harvard Medical School. Portions of this research were conducted on the O2 High Performance Compute Cluster, supported by the Research Computing Group, at Harvard Medical School. Fluorescence-activated cell sorting was performed by the Harvard Medical School Immunology Flow Cytometry Core Facility. This work was supported by NIH grants R01-DC015974, R01-DC009223 (LVG), and DIR-Z01-DC000091 (CJCW) along with a Blavatnik Sensory Disorders Research Grant (LVG). AAS was supported by NIH fellowship F32-DC019009 and the Harvard Mahoney Neuroscience Institute Fund. MCY was supported on a summer fellowship through the Harvard Amgen Scholars Program, which is funded by an Amgen Foundation Grant to Harvard University.

## Additional information

### Funding

| Funder | Grant reference number | Author |
| --- | --- | --- |
| National Institute on Deafness and Other Communication Disorders | R01-DC015974 | Lisa V Goodrich |
| National Institute on Deafness and Other Communication Disorders | R01-DC009223 | Lisa V Goodrich |
| NIH Office of the Director | Z01-DC000091 | Catherine JC Weisz |
| Blavatnik Family Foundation | Blavatnik Sensory Disorders Research Grant | Lisa V Goodrich |
| National Institute on Deafness and Other Communication Disorders | F32-DC019009 | Austen A Sitko |
| Harvard Mahoney Neuroscience Institute Fund | Postdoctoral Fellowship | Austen A Sitko |
| Amgen Foundation | Summer Fellowship | Mary Caroline Yuk |

The funders had no role in study design, data collection and interpretation, or the decision to submit the work for publication.

### Author contributions

Michelle M Frank, Conceptualization, Data curation, Formal analysis, Supervision, Validation, Investigation, Visualization, Methodology, Writing – original draft, Writing – review and editing; Austen A Sitko, Conceptualization, Data curation, Formal analysis, Supervision, Funding acquisition, Validation, Investigation, Visualization, Methodology, Writing – original draft, Writing – review and editing; Kirupa Suthakar, Lester Torres Cadenas, Investigation, Formal analysis, Writing – review and editing; Mackenzie Hunt, Investigation, Writing – review and editing, Formal analysis; Mary Caroline Yuk, Formal analysis, Writing – review and editing; Catherine JC Weisz, Formal analysis, Supervision, Writing – original draft, Writing – review and editing, Funding acquisition; Lisa V Goodrich, Conceptualization, Supervision, Funding acquisition, Writing – original draft, Project administration, Writing – review and editing

### Author ORCIDs

Michelle M Frank ⓘ http://orcid.org/0000-0002-6613-8251
Austen A Sitko ⓘ http://orcid.org/0000-0002-7601-6143
Catherine JC Weisz ⓘ http://orcid.org/0000-0002-2595-835X
Lisa V Goodrich ⓘ http://orcid.org/0000-0002-3331-8600

## Ethics

This study was performed in accordance with recommendations from the Guide for the Care and Use of Laboratory Animals. All experiments and procedures were approved by the Institutional Care and Use Committee of Harvard Medical School (protocol #IS00000067) or the National Institute on Deafness and Other Communication Disorders Animal Care and Use Committee. Every effort was made to minimize suffering throughout this work.

## Decision letter and Author response

Decision letter https://doi.org/10.7554/eLife.83855.sa1
Author response https://doi.org/10.7554/eLife.83855.sa2

## Additional files

### Supplementary files

• Supplementary file 1. Cell counts of NPY- and Ucn-expressing LOCs (related to *Figure 3*). LOCs were manually annotated as expressing high or low levels of NPY or Ucn. Cells where both peptides were expressed above background are considered to be co-expressing both peptides. Each row includes cell counts from a single, 40 µm-thick section.

• Supplementary file 2. Custom FISH probe sequences (related to *Figure 3*, *Figure 3—figure supplement 1*). Oligo probe sequences used in *Figure 3—figure supplement 1* for detecting *Penk* and *Pdyn* via hybridization chain reaction-based FISH.

• Supplementary file 3. Physiological properties of LOCs do not correlate with Calca-GFP fluorescence intensity (related to *Figure 6*). Results of single linear regression and correlation analysis between Calca-GFP and the indicated measures.

• Supplementary file 4. Results of statistical tests comparing peptide expression between OCN groups at different ages (related to *Figure 7*). Adjusted p-values from Kruskal-Wallis multiple-comparisons test for the indicated cell types and ages. Data is shown in *Figure 7A–D*.

• MDAR checklist

### Data availability

Single-cell data collected in this study is available on GEO, accession number GSE214027.

The following dataset was generated:

| Author(s) | Year | Dataset title | Dataset URL | Database and Identifier |
|---|---|---|---|---|
| Frank MM, Goodrich LV | 2022 | Single-nucleus sequencing data from developing and adult mouse olivocochlear neurons | https://www.ncbi.nlm.nih.gov/geo/query/acc.cgi?acc=GSE214027 | NCBI Gene Expression Omnibus, GSE214027 |

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

# Appendix 1

**Appendix 1—key resources table**

| Reagent type (species) or resource | Designation | Source or reference | Identifiers | Additional information |
|---|---|---|---|---|
| Strain, C57Bl/6J (*Mus musculus*) | *C57BL/6J* | The Jackson Laboratory | RRID:IMSR_JAX:000664 | |
| Genetic reagent (*Mus musculus*) | *Calca-GFP* | Gensat | RRID:MMRRC_011187-UCD | Full name: Tg(Calca-EGFP) FG104Gsat/Mmucd |
| Genetic reagent (*Mus musculus*) | *Rosa26^tdTomato* | The Jackson Laboratory | RRID:IMSR_JAX:007914 | Full name: B6.Cg-Gt(ROSA)26Sor^tm14(CAG-tdTomato)Hze/J |
| Genetic reagent (*Mus musculus*) | *Igs7^GFP* | The Jackson Laboratory | RRID:IMSR_JAX:030220 | Full name: B6.Cg-Igs7^tm140.1(tetO-EGFP,CAG-tTA2)Hze/J |
| Genetic reagent (*Mus musculus*) | *Rosa26^Sun1-GFP* | The Jackson Laboratory | RRID:IMSR_JAX:021039 | Full name: B6;129-Gt(ROSA)26Sor^tm5(CAG-Sun1/sfGFP)Nat/J |
| Genetic reagent (*Mus musculus*) | *Chat^Cre* | The Jackson Laboratory | RRID:IMSR_JAX:006410 | Full name: B6;129S6-Chat^tm2(cre)Lowl/J |
| Genetic reagent (*Mus musculus*) | *Chat^CreΔNeo* | The Jackson Laboratory | RRID:IMSR_JAX:031661 | Full name: B6.129S-Chat^tm1(cre)Lowl/MwarJ |
| Genetic reagent (*Mus musculus*) | *Npy^FlpO* | The Jackson Laboratory | RRID:IMSR_JAX:030211 | Full name: B6.Cg-Npy^tm1.1(flpo)Hze/J |
| Genetic reagent (*Mus musculus*) | *Ret^CreER* | David Ginty (Harvard); *Luo et al., 2009* | MGI:4437245 | Full name: Ret^tm2(cre/ERT2)Ddg |
| Genetic reagent (*Mus musculus*) | *Npy-GFP* | The Jackson Laboratory | RRID:IMSR_JAX:006417 | Full name: B6.FVB-Tg(Npy-hrGFP)1Lowl/J |
| Genetic reagent (*Mus musculus*) | *Tmc1^KO* | David Corey (Harvard); *Kawashima et al., 2011* | MGI:5427097 | Full name: Tmc1^tm1.1Ajg |
| Genetic reagent (*Mus musculus*) | *Tmc2^KO* | David Corey (Harvard); *Kawashima et al., 2011* | MGI:5427099 | Full name: Tmc2^tm1.1Ajg |
| Genetic reagent (*Mus musculus*) | *Rosa26^FLTG* | The Jackson Laboratory | RRID:IMSR_JAX:026932 | Full name: B6.Cg-Gt(ROSA)26Sor^tm1.3(CAG-tdTomato,-EGFP)Pjen/J |
| Antibody | Goat polyclonal Anti-Calb2 antiserum | Swant | Cat#CG1; RRID:AB_10000342 | (1:1,000) |
| Antibody | Goat polyclonal Anti-ChAT | Millipore Sigma | Cat#AB144P; RRID:AB_2079751; Lot 3675895 | (1:500) |
| Antibody | Mouse monoclonal Anti-CGRP | Abcam | Cat#ab81887; RRID:AB_1658411; Lots GR3336715-4, GR3374844-3, GR3233139-4 | (1:500–1:1,000) |

*Appendix 1 Continued on next page*

*Appendix 1 Continued*

| Reagent type (species) or resource | Designation | Source or reference | Identifiers | Additional information |
|---|---|---|---|---|
| Antibody | Chicken polyclonal Anti-GFP | Aves | Cat#GFP-1020; RRID: AB_10000240; Lot GFP3717982 | (1:2,000) |
| Antibody | Rabbit monoclonal Anti-NPY | Cell Signaling | Cat#11976; RRID: AB_2716286; Lots 3, 4, 5 | (Brainstem: 1:500) (Cochlea: 1:1,000) |
| Antibody | Guinea Pig polyclonal Anti-Parvalbumin | Synaptic Systems | Cat#195–004; RRID: AB_2156476; Lots 3–33, 3–36 | (1:1,000) |
| Antibody | Mouse monoclonal Anti-Synaptophysin 1 | Synaptic Systems | Cat#101 011; RRID: AB_887824; Lot 1–58 | (1:600) |
| Antibody | Chicken polyclonal Anti-TH | Abcam | Cat#ab76442; RRID: AB_1524535; Lot GR3393939-1 | (1:1,000) |
| Antibody | Rabbit polyclonal Anti-Ucn Polyclonal | Sigma-Alridch | Cat#SAB4503058; RRID:AB_10753229; Lot 310386 | (1:500) |
| Antibody | Donkey polyclonal Anti-Chicken 488 | Jackson ImmunoResearch | Cat#703-545-155; RID:AB_2340375 | (1:1,000) |
| Antibody | Donkey polyclonal Anti-Chicken 647 | Jackson ImmunoResearch | Cat#703-605-155; RRID:AB_2340379 | (1:1,000) |
| Antibody | Donkey polyclonal Anti-Goat 405 | Jackson ImmunoResearch | Cat#705-475-003; RRID:AB_2340426 | (1:1,000) |
| Antibody | Donkey polyclonal Anti-Goat 488 | Jackson ImmunoResearch | Cat#705-545-147; RRID:AB_2336933 | (1:1,000) |
| Antibody | Donkey polyclonal Anti-Goat 647 | ThermoFisher | Cat#A21447; RRID: AB_2535864 | (1:1,000) |
| Antibody | Donkey polyclonal Anti-Mouse 488 | Abcam | Cat#ab150105; RRID: AB_2732856 | (1:1,000) |
| Antibody | Donkey polyclonal Anti-Mouse 488 | Invitrogen | Cat#A21202; RRID: AB_141607 | (1:1,000) |
| Antibody | Donkey polyclonal Anti-Mouse 568 | ThermoFisher | Cat#A10037; RRID: AB_2534013 | (1:1,000) |
| Antibody | Donkey polyclonal Anti-Mouse 647 | ThermoFisher | Cat#A31571; RRID:AB_162542 | (1:1,000) |
| Antibody | Donkey polyclonal Anti-Rabbit 568 | ThermoFisher | Cat#A10042; RRID: AB_2534017 | (1:1,000) |
| Antibody | Donkey polyclonal Anti-Rabbit 647 | ThermoFisher | Cat#A31573; RRID:AB_2536183 | (1:1,000) |
| Antibody | Donkey polyclonal Anti-Guinea Pig 647 | Jackson ImmunoResearch | Cat#706–605–148; RRID:AB_2340476 | (1:1,000) |
| Sequence-based reagent | *Cadps2* FISH probe | Molecular Instruments | Custom probe | |
| Sequence-based reagent | *Calca* FISH probe | Molecular Instruments | *Calca* | |
| Sequence-based reagent | *Col4a4* FISH probe | Molecular Instruments | Custom probe | |

*Appendix 1 Continued on next page*

*Appendix 1 Continued*

| Reagent type (species) or resource | Designation | Source or reference | Identifiers | Additional information |
|---|---|---|---|---|
| Sequence-based reagent | *Zfp804a* FISH probe | Molecular Instruments | Custom probe | |
| Sequence-based reagent | *Penk* FISH probe | IDT | This paper | Sequence in *Supplementary file 2* |
| Sequence-based reagent | *Pdyn* FISH probe | IDT | This paper | Sequence in *Supplementary file 2* |
| Commercial assay or kit | 3' Single Cell Kit: 10x v2 | 10x Genomics | Cat#PN-120267 | |
| Commercial assay or kit | 3' Single Cell Kit: 10x v3 | 10x Genomics | Cat#PN-1000092 | |
| Commercial assay or kit | HCR RNA-FISH (Tissue section kit) | Molecular Instruments | N/A | |
| Commercial assay or kit | SPRIselect Reagent Kit | Beckman Coulter | Cat#B23318 | |
| Commercial assay or kit | DynaBeads MyOne Silane Beads | ThermoFisher | Cat#37002D | |
| Chemical compound, drug | 10% Tween | Bio-Rad | Cat#1610781 | |
| Chemical compound, drug | 10x PBS | ThermoFisher | Cat#70011044 | |
| Chemical compound, drug | 16% Paraformaldehyde | Electron Microscopy Sciences | Cat#15710 | |
| Chemical compound, drug | 1x PBS | ThermoFisher | Cat#10010031 | |
| Chemical compound, drug | 50% Glycerol | Ricca Chemical Company | Cat#3290–32 | |
| Chemical compound, drug | AlexaFluor-488 hydrazide | Fisher Scientific | Cat#A10436 | |
| Chemical compound, drug | APV | Tocris | Cat#0106 | |
| Chemical compound, drug | Biocytin | Sigma-Aldrich | Cat#B4261–25MG | |
| Chemical compound, drug | BSA | Sigma-Aldrich | Cat#A2058–5G | |
| Chemical compound, drug | BSA | ThermoFisher | Cat#BP1600–100 | |
| Chemical compound, drug | Buffer EB | Qiagen | Cat#19086 | |

*Appendix 1 Continued on next page*

*Appendix 1 Continued*

| Reagent type (species) or resource | Designation | Source or reference | Identifiers | Additional information |
|---|---|---|---|---|
| Chemical compound, drug | CaCl$_2$ | Sigma-Aldrich | Cat#C7902 | |
| Chemical compound, drug | Corn Oil | Sigma-Aldrich | Cat#C8267–500ml | |
| Chemical compound, drug | CsCl | Fisher Scientific | Cat#AC206320250 | |
| Chemical compound, drug | CsOH | Fisher Scientific | Cat#AC213601000 | |
| Chemical compound, drug | D-gluconic acid | Sigma-Aldrich | Cat#P1847–-100G | |
| Chemical compound, drug | DEPC (Diethyl Pyrocarbonate) | Sigma-Aldrich | Cat#D5758 | |
| Chemical compound, drug | Dextrose | Fisher Scientific | Cat#D16-500 | |
| Chemical compound, drug | DNQX | Sigma-Aldrich | Cat#D0540 | |
| Chemical compound, drug | DTT | Sigma-Aldrich | Cat#D0632 | |
| Chemical compound, drug | EGTA | Fisher Scientific | Cat#409910250 | |
| Chemical compound, drug | Ethanol | Decon Labs | Cat#V1016 | |
| Chemical compound, drug | Fluoromount-G | Southern Biotech | Cat#0100–01 | |
| Chemical compound, drug | Formalin | ThermoFisher | Cat#F79-500 | |
| Chemical compound, drug | Gelatin | Sigma-Aldrich | Cat#G1890–100G | |
| Chemical compound, drug | Glutaraldehyde | ThermoFisher | Cat#50–262–19 | |
| Chemical compound, drug | HEPES | Sigma-Aldrich | Cat#H3375 | |
| Chemical compound, drug | IGEPAL CA-630 | Sigma-Aldrich | Cat#I8896 | |

*Appendix 1 Continued on next page*

*Appendix 1 Continued*

| Reagent type (species) or resource | Designation | Source or reference | Identifiers | Additional information |
|---|---|---|---|---|
| Chemical compound, drug | Iodixanol, OptiPrep Density Medium | Sigma-Aldrich | Cat#D1556 | |
| Chemical compound, drug | KCl | Fisher Scientific | Cat#P217–500 | |
| Chemical compound, drug | KCl | Sigma-Aldrich | Cat#P9541–500G | |
| Chemical compound, drug | $KH_2PO_4$ | Fisher Scientific | Cat#P288-100 | |
| Chemical compound, drug | $KH_2PO_4$ | Sigma-Aldrich | Cat#P9791–500G | |
| Chemical compound, drug | KOH | Sigma-Aldrich | Cat#P4494-50ML | |
| Chemical compound, drug | Kynurenic acid | Sigma-Aldrich | Cat#K3375-5G | |
| Chemical compound, drug | Low TE Buffer | ThermoFisher | Cat#12090–015 | |
| Chemical compound, drug | Mg-ATP | Sigma-Aldrich | Cat#A9187 | |
| Chemical compound, drug | $MgCl_2$ | Sigma-Aldrich | Cat#M2670 | |
| Chemical compound, drug | $MgSO_4$ | Fisher Scientific | Cat#M65-500 | |
| Chemical compound, drug | Na-GTP | Sigma-Aldrich | Cat#G8877 | |
| Chemical compound, drug | $Na_2$-phosphocreatine | Sigma-Aldrich | Cat#P7936-10MG | |
| Chemical compound, drug | $Na_2HPO_4$ | Sigma-Aldrich | Cat#S3264-1KG | |
| Chemical compound, drug | NaCl | Fisher Scientific | Cat#S271–1 | |
| Chemical compound, drug | NaCl | Sigma-Aldrich | Cat#S3014-1KG | |
| Chemical compound, drug | $NaHCO_3$ | Sigma-Aldrich | Cat#S5761–500G | |

*Appendix 1 Continued on next page*

*Appendix 1 Continued*

| Reagent type (species) or resource | Designation | Source or reference | Identifiers | Additional information |
|---|---|---|---|---|
| Chemical compound, drug | NEG-50 | Epredia | Cat#6502 | |
| Chemical compound, drug | Normal Donkey Serum | Jackson ImmunoResearch | Cat#017–000–121 | |
| Chemical compound, drug | Nuclease-Free Water | ThermoFisher | Cat#AM9937 | |
| Chemical compound, drug | Protease inhibitor cocktail | Roche | Cat#11836170001 | |
| Chemical compound, drug | Proteinase K | Sigma-Aldrich | Cat#P-6556 | |
| Chemical compound, drug | QX-314 | Fisher Scientific | Cat#23–135–0 | |
| Chemical compound, drug | RNAsin Plus RNase Inhibitor | Promega | Cat#N2615 | |
| Chemical compound, drug | Sodium Citrate: $C_6H_5Na_3O_7 \cdot 2H_2O$ | Sigma-Aldrich | Cat#S4641–1KG | |
| Chemical compound, drug | Spermidine trihydrochloride | Sigma-Aldrich | Cat#S2501 | |
| Chemical compound, drug | Spermine tetrahydrochloride | Sigma-Aldrich | Cat#S1141 | |
| Chemical compound, drug | Sucrose | Sigma-Aldrich | Cat#S0389–500G | |
| Chemical compound, drug | Tamoxifen | Sigma-Aldrich | Cat#T5648–1G | |
| Chemical compound, drug | Tetrodotoxin (TTX) | Cayman Chemical | Cat#14964 | |
| Chemical compound, drug | Tricine | Sigma-Aldrich | Cat#T5816–100G | |
| Chemical compound, drug | Triton X-100 | Sigma-Aldrich | Cat#T9284 | |
| Chemical compound, drug | Tween 20 | Sigma-Aldrich | Cat#P9416–100ML | |
| Chemical compound, drug | β-estradiol | Sigma-Aldrich | Cat#E2758–1G | |

*Appendix 1 Continued on next page*

*Appendix 1 Continued*

| Reagent type (species) or resource | Designation | Source or reference | Identifiers | Additional information |
|---|---|---|---|---|
| Software, algorithm | MultiClamp 700B | Molecular devices | | |
| Software, algorithm | DigiData 1440 A | Molecular devices | | |
| Software, algorithm | Clampex 10.6 software | Molecular devices | | |
| Software, algorithm | Origin 2021 | OriginLabs | | |
| Software, algorithm | Adobe Illustrator 2021 | Adobe | | |
| Software, algorithm | Inkscape | inkscape.org | | |
| Software, algorithm | R | https://www.r-project.org/ | v.3.5.3–4.0.5 | |
| Software, algorithm | Seurat | *Stuart et al., 2019* | v.3.1.4–3.2.3 | |
| Software, algorithm | Imaris | Bitplane AG | v.9.8–9.9.0 | |
| Software, algorithm | MATLAB | MathWorks | R2020a, R2021b | |
| Software, algorithm | cellranger | 10x Genomics | v.2.1.0–3.0.2 | |
| Software, algorithm | CellPose | *Stringer et al., 2021* | v.1 | |
| Software, algorithm | Prism | GraphPad | v.9.3.1 | |
| Software, algorithm | Zeiss ZEN | Zeiss | 2.3 (blue edition) | |
| Software, algorithm | Image processing: Local Normalization | *Zhang et al., 2021* | https://github.com/xzhang03/Local-normalize | |
| Software, algorithm | ImageJ | National Institutes of Health | | |
| Software, algorithm | Measure_Line ImageJ Plugin | Eaton Peabody Labs, MEEI | https://meeeplfiles.partners.org/Measure_line.class | |
| Other | Brain matrix | Zivic Instruments | Cat#BSMNS001–1 | See Methods, "Single-nucleus sequencing" |
| Other | FlowMi cell strainer | Sigma | Cat#BAH136800040 | See Methods, "Single-nucleus sequencing" |
| Other | DAPI | Invitrogen | Cat#D1306 | DNA dye, (1:10,000) See Methods, "Single-nucleus sequencing" & "Immunofluorescence" |
| Other | Draq7 | Abcam | Cat#AB109202 | DNA dye, (1:100) See Methods, "Single-nucleus sequencing" |

