## [Editor Report]

This paper provides a detailed cellular and molecular characterization of the olivocochlear efferents that project to the inner ear. These specialized motoneurons are the only source of feedback from the brain to the ear and have been difficult to access. This study comprehensively categorizes the efferents, using single nucleus RNA-sequencing and 3D reconstructions of individual fibers and their pre-synaptic contacts onto target neurons in the cochlea.

---

## [Decision Letter]

**Decision letter after peer review:**

Thank you for submitting your article "Experience-dependent flexibility in a molecularly diverse central-to-peripheral auditory feedback system" for consideration by *eLife*. Your article has been reviewed by 3 peer reviewers, including Catherine Emily Carr as Reviewing Editor and Reviewer #1, and the evaluation has been overseen by Barbara Shinn-Cunningham as the Senior Editor. The following individual involved in the review of your submission has agreed to reveal their identity: Zhiyong Liu (Reviewer #2).

Essential revisions:

1) Please modify your title, along the lines of perhaps along the lines of " Single cell transcriptomic profiling reveals xxxxx"

2) Follow the reviewer's suggestions to improve clarity.

*Reviewer #2 (Recommendations for the authors):*

Frank et al. in this study thoroughly characterized the molecular, physiological, and morphological properties of the brainstem olivocochlear neurons (OCNs). First, by using the ChATCre+; Rosa26-Sun1-GFP/+ and 10x single cell RNA-Seq platform, they identified two Gata3+ cell clusters that are defined as OCNs: one is Ucn+ LOC, and the other is Atp1a3+ MOC. By further comparing the gene profiles between OCNs and the lineage-related FMNs, several previously unknown OCN markers such as Cadps2 are revealed. Furthermore, they also noticed that OCNs have high internal heterogeneity and they further found LOC and MOC-enriched genes, including but not limited to Col4a4 and Zfp804a, respectively. More importantly, two LOC subtypes, LOC-1 and LOC-2, are identified, and especially the LOC-2 sub-cluster cells, which are located in the medial wing of the LSO, can enrich many neuropeptide genes, CGRP, Calcb, NPY, Ucn and so on. The NPY expression in LOC-2 is further validated by the well-designed genetic fate mapping approach, which is convincing. Finally, the author showed that the neuropeptide expression patterns did not significantly change between wild-type and Tmc1/2 knockout mice at the pre-hearing ages, but either the neuropeptide expression levels or distribution patterns did change manifestly after hearing onset. Besides, they also showed that noise exposure can alter the neuropeptide patterns.

In general, the paper is clearly written and the data are well presented. The logic is straightforward, and in most cases, the data are compelling to support the conclusions. I only have 2 main comments below:

1) In my point of view, it should be a resource paper rather than a research article because the current version did not connect any specific marker pattern with the biological properties. For example, the Npy identity does not appear to be predictive of LOC morphology. It remains unknown whether the neuropeptide changes are just a final outcome or really an important cellular event. Convincing evidence is lacking to address this question.

2) The author claimed that opioid (such as Pdyn and Penk) expression is not detected. I feel the authors tend to say it is a surprising observation. In my mind, it needs more caution, and it is better to confirm this negative observation by fate mapping analysis with Pdyn-cre (it is available). Fate mapping analysis is more solid to make a negative conclusion especially when we do not know the exact gene expression patterns. The samples should be analyzed at adult ages.

*Reviewer #3 (Recommendations for the authors):*

I strongly support publication in *eLife* given the indicated revisions.

I would like to congratulate the authors on this great study! Below please find some suggestions that might be helpful to further improve this manuscript.

Abstract:

"through unclear mechanisms" is inappropriate here given the amount of work and evidence on candidate mechanisms. This needs to be changed/weakened.

"we discovered a neuropeptide-enriched LOC subtype" given that LOCs have generally been thought to use neuropeptides and the analysis, as is, focuses on RNAseq I found this statement very strong (too strong), if not backed by immunohistochemistry in the cochlea.

Introduction:

There are some issues with references, which I tried to help with below. In general, according to DORA it would be preferred to cite original publications rather than reviews. However, there are cases that are well served by reviews: such as outer hair cell function.

Yet, the review of van der Heijden and Vavakou actually questions the cycle-by-cycle amplification by outer hair cells (OHCs) based on their OCT recordings (Vavakou et al., 2021). Perhaps cite a review of Dallos or others on electromotility and reference van der Heijden and Vavakou in the kind of (but see..) manner. Moreover, Glowatzki and Fuchs 2001 is on inner hair cells rather than OHCs. Also, the choice of the two Oatman papers might not be so helpful. Finally, the phrase "the detection of signals in noise26-28 and attention29-32." might be reconsidered (e.g. "effects of attention"?).

"No specific role in hearing has been definitively linked to LOCs, although they appear 78 to protect the ear from noise damage through unknown mechanisms34,35." While the shortage of functional data certainly applies, this statement and the references are falling quite short. I think you should tone down and consider citing Ruel et al., 2004 as well as Darrow et al., 2006.

Results:

"we adopted a single-nucleus sequencing strategy" While the approach will be clear to most of us it might still help clarify that this means cell nuclei rather than brainstem nuclei.

"Cell types were also similar in the fraction of mitochondrial genes detected," maybe I am missing something, but since there is mitochondrial DNA and nuclear sequencing was applied this should probably specify "nuclear-encoded mitochondrial genes" or the like. Moreover, the link between similarity in n-mt genes and homogeneous cell health might not be so obvious without a reference.

"OCNs but not in other motor neuron" this and later reads like you say OCNs are motor neurons: perhaps clarify in the intro that OCNs and "motor neurons" have common progenitors?

"Na,K-ATPase Atp1a3, which is selectively expressed in MOCs in mature rats45".

As shown by McLean et al., 2009 and other studies, in the cochlea, ATP1a3 is primarily expressed in type I SGNs, so this statement needs to be reconsidered.

"CADPS2 is linked to exocytosis of dense-core vesicles47-49",

I suggest mentioning the synonymous name CAPS-2, which is more frequently used in synaptic neuroscience, and to put its function into the context of vesicle priming (e.g. " a member of the Munc13 and CAPS family of priming proteins") potentially also quoting relevant neuronal work such as Jockusch et al., 2007, Vogl et al., 2015, Nestvogel et al., 2020.

Figure 1L and M: I found it difficult to detect any green signal in the respective left panels (also applies to other FISH analyses).

"Consistent with their integration into fundamentally different circuitry, OCNs and FMNs also differ in the expression of molecules relevant to mature function (Figures S2F, I). In particular, only OCNs express Gad2, a gene involved in GABA synthesis, whereas only FMNs express the serotonin receptor gene Htr2c (Figure S2I)." This is only one of the instances where the question of what predictive power RNA analysis has for protein expression is relevant. There is nothing wrong with the statement in principle, but is seems overtly simple to conclude this.

Figure 2, legend: "gene Zfp804a is expressed in MOCs (G) but is absent from most LOCs (F)."

"Both MOCs and LOCs have the ability to respond to numerous neurotransmitters, including 172 GABA (K), Glycine (L), acetylcholine (M), and glutamate (O)." another instance where care should be taken not to overinterpret the data.

Figure 3

It would be helpful to quantitatively see correlations between CGRP, NPY, and Ucn expression levels.

Figure 4

Please provide more explanation for how the masking of synaptophysin immunofluorescence was done.

"Strikingly, some (shown in Figure 4A-A"), but not all (Figure S5F-I), of these terminal fibers contained Syp puncta in the ISB." What precisely did you find striking here?

Please introduce CALB2 (calretinin), and also introduce the possibility of LOC-IHC contacts (e.g. Liberman et al., 1990, Lauer et al., 2012), which could confound the on-CALB2 analysis.

Figure 5

It looks like the majority of fibers that were analyzed have a considerable amount of NPY-positive terminals. However, the LOC2 cluster size seems to be smaller compared to LOC1, both from RNA sequencing data and qualitative immunolabeling. Is it possible that it has to do with Ret expression in different LOC subtypes? Please comment on how you reconcile these findings.

The section in Figure 5P is quite dense and a bit cryptic. I generally appreciate the effort to quantify potential relationships as done here but then again worry about how conclusive this is.

Legend to Figure 5: "High CALB2 fibers correspond to low threshold Type Ia SGNs; low CALB2 fibers correspond to high threshold Type Ic SGNs." To my knowledge, the correspondence of the molecular profiles and functional properties has yet to be tested. Please rephrase.

"….directly contacted the CALB2 surface ("On CALB2" in magenta) or not ("Off CALB2" in green).."

"…oriented with cochlear base to.."

The authors claim that there is no distinct pattern of NPY positive terminal connectivity along the pillar-modiolar axis. It seems that this statement is based on qualitative observations and the percentages of NPY+ terminals occurring on Syp+ puncta that are on or off CALB2. The authors could support this claim further, for instance, by a plot showing how many NPY+ Syp puncta occur on the pillar vs modiolar side.

Physiological characterization:

If analysis focused on the medial wing of LSO: how would you have a chance to pick up LOC1 given the data provided earlier in the manuscript?

When/how was the GFP signal recorded: prior to break-in (avoid loss via diffusion into the pipette?)?

Discussion:

"Molecular LOC subtypes are morphologically and physiologically diverse" I don't think the claim of physiological diversity is sufficiently supported by the data. Consider removing it from the section title and mark as important speculation and objective for future studies.

---

## [Author Response]

Reviewer #2 (Recommendations for the authors):Frank et al. in this study thoroughly characterized the molecular, physiological, and morphological properties of the brainstem olivocochlear neurons (OCNs). First, by using the ChATCre+; Rosa26-Sun1-GFP/+ and 10x single cell RNA-Seq platform, they identified two Gata3+ cell clusters that are defined as OCNs: one is Ucn+ LOC, and the other is Atp1a3+ MOC. By further comparing the gene profiles between OCNs and the lineage-related FMNs, several previously unknown OCN markers such as Cadps2 are revealed. Furthermore, they also noticed that OCNs have high internal heterogeneity and they further found LOC and MOC-enriched genes, including but not limited to Col4a4 and Zfp804a, respectively. More importantly, two LOC subtypes, LOC-1 and LOC-2, are identified, and especially the LOC-2 sub-cluster cells, which are located in the medial wing of the LSO, can enrich many neuropeptide genes, CGRP, Calcb, NPY, Ucn and so on. The NPY expression in LOC-2 is further validated by the well-designed genetic fate mapping approach, which is convincing. Finally, the author showed that the neuropeptide expression patterns did not significantly change between wild-type and Tmc1/2 knockout mice at the pre-hearing ages, but either the neuropeptide expression levels or distribution patterns did change manifestly after hearing onset. Besides, they also showed that noise exposure can alter the neuropeptide patterns.In general, the paper is clearly written and the data are well presented. The logic is straightforward, and in most cases, the data are compelling to support the conclusions.

Thank you for the kind comments.

I only have 2 main comments below:1) In my point of view, it should be a resource paper rather than a research article because the current version did not connect any specific marker pattern with the biological properties. For example, the Npy identity does not appear to be predictive of LOC morphology. It remains unknown whether the neuropeptide changes are just a final outcome or really an important cellular event. Convincing evidence is lacking to address this question.

We agree with the reviewer that the significance of NPY expression remains to be determined. However, the research finding of note is the discovery of a population of LOCs marked by NPY – these NPY-positive LOCs are molecularly distinct and anatomically segregated, but they do not exhibit stereotyped morphologies or patterns of connectivity. We provide multiple types of evidence supporting this conclusion, including single nucleus RNA-seq data, immunohistochemistry, genetic fate mapping, and 3D reconstructions of individual axonal arbors in the cochlea. Moreover, a growing body of literature suggests that cellular subtype identity is rarely predictive of cellular morphology, as described in the Discussion (see e.g. Kim, …, Anderson 2019; Peng, …, Zeng 2021; general lack of correlation between transcriptional subtypes and morphological characteristics reviewed in Zeng 2022, *Cell*). Thus, we feel that the lack of correlation between NPY expression and cellular morphology is a biologically relevant observation that is best shared in the form of a research article. To make this point more clearly, we expanded the discussion of LOC heterogeneity (pg. 11, lines 504-514).

With respect to the role of NPY itself, we feel that the data do indeed provide compelling evidence that NPY (and other peptides) are induced by acoustic trauma, a finding that impacts how we think about LOC function more broadly, as presented in the Discussion (p. 20-21, “Functional consequences of dynamic neuropeptide expression”). Our work on neuropeptide induction is consistent with work in other peripheral systems, including neurons innervating the skin (Hoeffel, …, Ugolini 2021) and gut (Yang, …, Chiu 2022; see Discussion). Given the consistency of these phenotypes, the effect size (~4-5x increase in Ucn- or NPY-expressing LOCs, p<0.001), and agreement with the available literature in other sensory systems, it is reasonable to conclude that peptide induction in olivocochlear neurons is a genuine phenomenon that could play a similar role to that seen elsewhere in the body. Investigating the impact of this peptide induction is an important goal for the future, but is beyond the scope of the current study. This point has been added to the Discussion (pg. 13, lines 589-591).

2) The author claimed that opioid (such as Pdyn and Penk) expression is not detected. I feel the authors tend to say it is a surprising observation. In my mind, it needs more caution, and it is better to confirm this negative observation by fate mapping analysis with Pdyn-cre (it is available). Fate mapping analysis is more solid to make a negative conclusion especially when we do not know the exact gene expression patterns. The samples should be analyzed at adult ages.

The reviewer is correct that a negative result detecting *Pdyn* and *Penk* transcription is not conclusive evidence that LOC neurons never use these peptides. Although we include images of *Pdyn*- and *Penk*-expressing positive control cells, we did not definitively image the entire LOC across sections, so we cannot rule out the possibility that we may have missed expression in a subset of LOCs. We have adjusted our language in the text accordingly (pg. 5, lines 201-202). We agree that fate mapping using *Pdyn-Cre* could be interesting, but these experiments would not address the central question of whether LOC neurons signal with opioid peptides in adult animals, as there are several circuits where neurons have a history of neurotransmitter expression that is not retained in adults, including cells in the auditory system (reviewed in Spitzer 2017).

Reviewer #3 (Recommendations for the authors):I strongly support publication in eLife given the indicated revisions.I would like to congratulate the authors on this great study! Below please find some suggestions that might be helpful to further improve this manuscript.Abstract:"through unclear mechanisms" is inappropriate here given the amount of work and evidence on candidate mechanisms. This needs to be changed/weakened.

We agree and have removed the phrase accordingly (pg. 1).

"we discovered a neuropeptide-enriched LOC subtype" given that LOCs have generally been thought to use neuropeptides and the analysis, as is, focuses on RNAseq I found this statement very strong (too strong), if not backed by immunohistochemistry in the cochlea.

We appreciate the reviewer’s caution in drawing strong conclusions about the discovery of an LOC subtype. Examining expression of proteins other than NPY (Figure 5 and its supplement) or CGRP (e.g. Wu et al. 2017) in the cochlea would be an interesting and worthwhile set of experiments that could inform our understanding of how peptide release might influence cochlear function. However, we feel that our current approach is better suited to support the claim we make here, namely that neuropeptide-enriched LOCs exist. We focused on the cell bodies of LOCs in the brainstem as that is the only location in which we can reliably distinguish individual cells, given the extensive branching of their axons in the ear. The brainstem is therefore a more reliable place to look for variability in peptide expression between individual LOC neurons than the cochlea. In addition, we based our conclusion that there is a neuropeptide-enriched LOC subtype not only on RNAseq (Figure 3), but also on quantitative immunohistochemistry (Figure 3, 7, 7—figure supplement 1, and 8), *in situ* hybridization (Figure 3), fate mapping (Figure 3), and expression of promoter-based fluorescent reporter lines for NPY and CGRP (Figure 3, 6). All of these methods yield consistent results, indicating that bias in peptide expression arises in a stereotyped manner at the level of promoters, transcripts, and protein. Finally, although LOCs have indeed long been thought to signal through peptides, earlier work has not thoroughly examined the possibility of distinct cohorts of LOCs with varying peptide expression. Nonetheless, prior work in other species does recapitulate our finding of a medial bias in peptide expression across the LSO at the level of protein expression (Kaiser et al. 2011, Vetter et al. 1991). Thus, although additional immunohistochemistry in the cochlea would certainly be of interest, those results would not affect our central conclusion of variability in peptide expression across individual LOCs. We have added a sentence to the Discussion to better explain the evidence for a neuropeptide-enriched LOC subtype (pg. 11, lines 504-514).

Introduction:There are some issues with references, which I tried to help with below. In general, according to DORA it would be preferred to cite original publications rather than reviews. However, there are cases that are well served by reviews: such as outer hair cell function.Yet, the review of van der Heijden and Vavakou actually questions the cycle-by-cycle amplification by outer hair cells (OHCs) based on their OCT recordings (Vavakou et al., 2021). Perhaps cite a review of Dallos or others on electromotility and reference van der Heijden and Vavakou in the kind of (but see..) manner. Moreover, Glowatzki and Fuchs 2001 is on inner hair cells rather than OHCs. Also, the choice of the two Oatman papers might not be so helpful. Finally, the phrase "the detection of signals in noise26-28 and attention29-32." might be reconsidered (e.g. "effects of attention"?).

Thank you for your careful attention to detail in checking these references! We have added an additional review on outer hair cell electromotility (pg. 2, line 53) and replaced the Glowatzki and Fuchs 2001 reference with papers discussing the OHC-MOC synapse (Blanchet et al. 1996 and Dallos et al. 1997; pg. 2, line 54). We also revised the phrasing in line 56-57. Changes to the references are highlighted in yellow.

"No specific role in hearing has been definitively linked to LOCs, although they appear 78 to protect the ear from noise damage through unknown mechanisms34,35." While the shortage of functional data certainly applies, this statement and the references are falling quite short. I think you should tone down and consider citing Ruel et al., 2004 as well as Darrow et al., 2006.

Thank you for the suggestion. We have rephrased this sentence and added references to Ruel et al. 2001 and Wu et al. 2020 (concerning the role of dopamine in tuning SGN excitability) and Darrow et al. 2006, Larsen and Liberman 2010, and Irving et al. 2011 (concerning the possibility of OCNs in sound localization circuitry). We also added a reference to an additional review article (Fuente et al. 2015) describing possible mechanisms mediating LOC protection from acoustic trauma for readers interested in exploring the topic beyond the scope of this paper (pg. 2, lines 69-74). Changes to the references are highlighted in yellow.

Results:"we adopted a single-nucleus sequencing strategy" While the approach will be clear to most of us it might still help clarify that this means cell nuclei rather than brainstem nuclei.

We have rewritten the text for clarity (pg. 2, lines 87-88).

"Cell types were also similar in the fraction of mitochondrial genes detected," maybe I am missing something, but since there is mitochondrial DNA and nuclear sequencing was applied this should probably specify "nuclear-encoded mitochondrial genes" or the like. Moreover, the link between similarity in n-mt genes and homogeneous cell health might not be so obvious without a reference.

We appreciate the reviewer’s careful attention to important analytic details. The fraction of mitochondrial genes detected is a standard metric to assess the quality of single-cell or single-nucleus sequencing data. This analysis specifically includes genes mapping to the mitochondrial genome, not nuclear-encoded mitochondrial genes. As described in the Methods, the presence of genes mapping to the mitochondrial genome likely indicates contaminants from outside the nucleus or other issues with sample quality. We have revised this phrasing for clarity and added appropriate references in both the Results and the Methods (pg. 3, line 108-109; pg. 15, lines 675-679). Changes to the references are highlighted in yellow.

"OCNs but not in other motor neuron" this and later reads like you say OCNs are motor neurons: perhaps clarify in the intro that OCNs and "motor neurons" have common progenitors?

This is a helpful suggestion, and we have added this information to the section of the Introduction that discussed the motor neuron origin of OCNs (pg. 1, lines 39-41).

"Na,K-ATPase Atp1a3, which is selectively expressed in MOCs in mature rats45".As shown by McLean et al., 2009 and other studies, in the cochlea, ATP1a3 is primarily expressed in type I SGNs, so this statement needs to be reconsidered.

We agree that the use of the word “selectively” requires more explanation. We were referring specifically to preferential ATP1a3 expression in OCN cell bodies, so additional expression in SGNs or other cell types does not affect its utility as a distinguishing marker between MOCs and LOCs. We have updated this phrasing for clarity (pg. 3, line 114).

"CADPS2 is linked to exocytosis of dense-core vesicles47-49",I suggest mentioning the synonymous name CAPS-2, which is more frequently used in synaptic neuroscience, and to put its function into the context of vesicle priming (e.g. " a member of the Munc13 and CAPS family of priming proteins") potentially also quoting relevant neuronal work such as Jockusch et al., 2007, Vogl et al., 2015, Nestvogel et al., 2020.

This is a great point. We adopted this phrasing and added additional references related to CAPS function (pg. 3, lines 125-127). Changes to the references are highlighted in yellow.

Figure 1L and M: I found it difficult to detect any green signal in the respective left panels (also applies to other FISH analyses).

We agree that it can be difficult to interpret the FISH analyses. We have tried our best to choose colors that are visible in the overlay. However, given the size of the puncta, there are limits to how much we can do to make the green points visible. For this reason, we also show an inverted image of the single-channel data with the mRNA puncta labeled in black. We feel that this is the best way to represent the data. We have also made slight adjustments to the overlay images shown in Figures 1L, 2C, and 2G to make the overlapping signals easier to see. There is no green signal visible in Figure 1M because the *Cadps2* transcript is not expressed in the facial nucleus, which is more clearly seen in the single-channel panel on the right of 1M.

"Consistent with their integration into fundamentally different circuitry, OCNs and FMNs also differ in the expression of molecules relevant to mature function (Figures S2F, I). In particular, only OCNs express Gad2, a gene involved in GABA synthesis, whereas only FMNs express the serotonin receptor gene Htr2c (Figure S2I)." This is only one of the instances where the question of what predictive power RNA analysis has for protein expression is relevant. There is nothing wrong with the statement in principle, but is seems overtly simple to conclude this.Figure 2, legend: "gene Zfp804a is expressed in MOCs (G) but is absent from most LOCs (F).""Both MOCs and LOCs have the ability to respond to numerous neurotransmitters, including 172 GABA (K), Glycine (L), acetylcholine (M), and glutamate (O)." another instance where care should be taken not to overinterpret the data.

We did not mean to imply that RNA analyses perfectly predict protein expression or function. We have made a number of changes throughout the text to better indicate that we are referring to genes. We also added a sentence to the Discussion indicating that additional confirmation is needed (pg. 12, lines 533-535).

Figure 3It would be helpful to quantitatively see correlations between CGRP, NPY, and Ucn expression levels.

We agree that this would be a useful analysis. However, given the limitations of single-cell sequencing data (and in particular the v2 10x kit we used for collecting the adult nuclei in this study), we do not think these results are reliable enough to include here. Specifically, droplet-based single-cell sequencing approaches are known to have issues with dropouts and false zero counts due to stochasticity in transcript detection, an effect that is more pronounced for kits with older chemistries, such as the 10x v2 kit (see e.g. Hicks, …, Irazarry 2017). Thus, although Ucn and NPY expression in adult LOCs is tightly correlated with an adjusted *R^2^* of 0.91 for cells in which both Ucn and NPY were detected, this analysis only includes 5 cells. Droplet-based sequencing platforms (like 10x) are also limited in their ability to detect the dynamic range of transcript expression, especially for lowly expressed transcripts, like peptides (e.g. Wang, … Zhang 2021), making correlation analyses based on transcript expression levels even more unreliable. We therefore used an orthogonal method to qualitatively examine co-expression of peptides at the protein and promoter levels, as shown in Figure 3N, O and Supplementary File 1. Because commercially available antibodies for both NPY and Ucn are raised in rabbit, we were unable to quantitatively compare NPY and Ucn protein expression in the same cells.

Figure 4Please provide more explanation for how the masking of synaptophysin immunofluorescence was done.

Additional details have been added to the Methods (pg. 18, lines 804-808).

"Strikingly, some (shown in Figure 4A-A"), but not all (Figure S5F-I), of these terminal fibers contained Syp puncta in the ISB." What precisely did you find striking here?

What stood out to us is that there do not appear to be two types of MOCs – those that form putative synapses with Type I SGN peripheral fibers and those that do not. Instead, individual MOC axons can extend some terminal branches that form synaptophysin puncta in the ISB while other terminal branches from the same axon bypass the ISB and exclusively terminate on OHCs. We edited the text to better communicate what is interesting about this observation (pg. 6, lines 250-252).

Please introduce CALB2 (calretinin), and also introduce the possibility of LOC-IHC contacts (e.g. Liberman et al., 1990, Lauer et al., 2012), which could confound the on-CALB2 analysis.

Although CALB2 is a reliable way to distinguish Ia vs Ib/c SGN subtypes (Shrestha…Goodrich, 2018), we agree that its expression in IHCs can be problematic. To make this point more clearly, we have re-written the text about why CALB2 was used and to acknowledge that we cannot rule out the possibility of LOC-IHC contacts. If such contacts exist, though, they are likely to be a minority, as most Syp puncta were well away from the IHCs. In the revised text, CALB2 is still introduced as a marker for both SGNs and IHCs when we describe the overall approach on pg. 5 (lines 224-227) and p. 6 (lines 262-268), and then we expanded the discussion of possible caveats on pg. 7, lines 293-301.

Figure 5It looks like the majority of fibers that were analyzed have a considerable amount of NPY-positive terminals. However, the LOC2 cluster size seems to be smaller compared to LOC1, both from RNA sequencing data and qualitative immunolabeling. Is it possible that it has to do with Ret expression in different LOC subtypes? Please comment on how you reconcile these findings.

The reviewer raises an interesting and important point. Indeed, LOC axons with high NPY content are likely over-represented in our morphological analyses. It is possible that our tamoxifen dosing scheme may have enriched for LOC neurons with medium-to-high NPY load in their axons. In adult animals, *Ret* expression does not appear to be higher in LOC2s than LOC1s. (*Ret* was detected in 24% of LOC1s and 20% of LOC2s, with an average normalized expression of 2.1 and 1.9 in *Ret*+ LOC1s and LOC2s, respectively. Note that the low percentage of cells with *Ret* detected likely reflects limitations of droplet-based single-cell sequencing methods, and does not imply that only ~20% of mature LOCs are producing *Ret* transcripts). When we give high doses of tamoxifen to mature (~P20–P28) *Ret^CreER^;Igs7^GFP^* mice, MOCs and LOCs are labeled extensively throughout the base-to-apex extent of the cochlea. However, we found that we were better able to reliably titrate sparse labeling in the *Ret^CreER^* line with embryonic tamoxifen doses than with postnatal or adult administration. It is possible that administering tamoxifen during late embryogenesis captured a developmentally distinct subpopulation of LOCs. Because many components of the auditory system develop from high-to-low frequencies, our embryonic injections could have captured an LOC population biased to the medial LSO, which would preferentially target NPY-high LOCs.

However, we feel confident in our experimental approach given the variety of LOC axonal morphologies we found as well as the large number of LOC axons with low or no NPY load. Our goal was to determine whether fibers with high NPY expression exhibited stereotyped morphologies in the cochlea, and the range of morphologies we identified indicates that they do not. Similarly, we found that axons with 0% NPY/Syp+ puncta do not exhibit any stereotyped properties, reinforcing our conclusion that peptide expression does not dictate axonal morphology. We have added a summary of our reasoning as well as potential caveats to the text (pg. 11, lines 504-514).

The section in Figure 5P is quite dense and a bit cryptic. I generally appreciate the effort to quantify potential relationships as done here but then again worry about how conclusive this is.

We agree that this section contains a lot of information that can be hard to digest. The point we are trying to make is that peripheral axon morphological features do not correlate with peptidergic content, which requires walking through a wide variety of parameters. We have rewritten this entire section to make the main point clear – namely, that LOCs are a diverse population of neurons containing two molecularly distinct subtypes, but whose innervation patterns in the cochlea suggest that they are poised to provide widespread and broad feedback to the cochlea. These changes are throughout the section entitled “NPY-positive LOC axons exhibit highly variable patterns of connectivity”, starting on p. 6.

Legend to Figure 5: "High CALB2 fibers correspond to low threshold Type Ia SGNs; low CALB2 fibers correspond to high threshold Type Ic SGNs." To my knowledge, the correspondence of the molecular profiles and functional properties has yet to be tested. Please rephrase.

We apologize for the confusing use of the word “correspond”, which is appropriate for the molecular subtypes but not for their functional properties. We have removed this sentence from the figure caption (pg. 30-31).

"….directly contacted the CALB2 surface ("On CALB2" in magenta) or not ("Off CALB2" in green)..""…oriented with cochlear base to.."

We have made both of these wording changes to the figure legend (pg. 30-31).

The authors claim that there is no distinct pattern of NPY positive terminal connectivity along the pillar-modiolar axis. It seems that this statement is based on qualitative observations and the percentages of NPY+ terminals occurring on Syp+ puncta that are on or off CALB2. The authors could support this claim further, for instance, by a plot showing how many NPY+ Syp puncta occur on the pillar vs modiolar side.

As the reviewer suggests, we used the On and Off CALB2 contact as a proxy for pillar and modiolar position of Syp puncta (with or without NPY). Although correlative, this is a reasonable readout of modiolar-pillar position that adequately serves the purpose of our analysis (i.e., to determine if there are clear biases in the localization of peptidergic or non-peptidergic LOC Syp puncta). The stereotypy of CALB2 expression gradients among the SGN peripheral processes along the modiolar-pillar axis of the base of the IHC have been shown previously by multiple groups (e.g. Shrestha…Goodrich, 2018; Sun…Mueller, 2018) and, as illustrated in the orthogonal view in Figure 5E, we found the alignment between modiolar/pillar position and On and Off CALB2 to be quite reliable. While we agree it would be better to directly assess position, it would be a considerable amount of additional work to code the modiolar/pillar position of each Syp puncta in Imaris. Given the lack of qualitative evidence that there are ever any CALB2-on Syp+ puncta on the modiolar side, we feel it would not substantially add meaningful insight to our conclusions about the lack of an anatomical bias in the position of NPY+ Syp+ puncta. To make this logic clear, we have rewritten the text in this section (pg. 6, lines 262-268, 270-271; pg. 7, lines 293-301).

Physiological characterization:If analysis focused on the medial wing of LSO: how would you have a chance to pick up LOC1 given the data provided earlier in the manuscript?

Although peptide-enriched LOC2s are restricted to the medial wing, cells with variable levels of CGRP-GFP are also present in the medial half of the LSO **(**Figure 6A’), consistent with the presence of both NPY+ and NPY- LOCs in this region (Figure S5C). We therefore chose to focus our analysis on cells in the mid-central LSO so that we could capture a range of CGRP intensities while limiting potential confounds associated with other types of variability along the length of the LSO (especially well-documented tonotopic variability). We have updated the text to make this logic clearer (pg. 8, lines 346-350).

When/how was the GFP signal recorded: prior to break-in (avoid loss via diffusion into the pipette?)?

GFP signal was measured prior to breaking into the cell. We have updated this detail in the Methods (pg. 20, lines 899-901).

Discussion:"Molecular LOC subtypes are morphologically and physiologically diverse" I don't think the claim of physiological diversity is sufficiently supported by the data. Consider removing it from the section title and mark as important speculation and objective for future studies.

Although we did not map LOC subtypes to specific physiological properties, we did find a range of physiological properties in the LOC neurons we recorded from (Figure 6). However, we agree that the initial heading implies that there are physiological subtypes of LOC neurons that correspond to molecular subtypes, which is not reflected in our data. We have updated this header accordingly (pg. 11, line 469).